# OPTIMIZATION ON MANIFOLDS WITH RIEMANNIAN JACOBIAN REGULARIZATION

## ABSTRACT

Understanding the effectiveness of intrinsic geometry in enhancing a model's generalization ability, we draw upon prior works that apply geometric principles to optimization and present a novel approach to improve robustness and generalization for constrained optimization problems. This work aims to strengthen the sharpness-aware optimizers and proposes a novel Riemannian optimizer. We first present a theoretical analysis that characterizes the relationship between the general loss and the perturbation of the empirical loss in the context of Riemannian manifolds. Motivated by the result obtained from this analysis, we introduce our algorithm named Riemannian Jacobian Regularization (RJR), which explicitly regularizes the Riemannian gradient norm and the projected Hessian. To demonstrate RJR's ability to enhance generalization, we evaluate and contrast our algorithm on a broad set of problems, such as image classification and contrastive learning across different datasets with various architectures.

## 1 INTRODUCTION

In deep learning and statistics, overfitting is a long-standing and challenging problem in which the model fails to generalize to the whole population due to the training process getting stuck in one of the local minima of the landscape of loss functions. This is attributed to high-dimensional and non-convex loss functions, which have a complicated landscape with multiple local minima. Regarding this issue, flat minimizers that seek regions with low sharpness have been known to be among the most effective approaches (Keskar et al., 2016; Kaddour et al., 2022b; Li et al., 2022). Sharpness-aware minimization (SAM), as introduced by Foret et al. (2021b), stands out as a notable method by simultaneously minimizing the loss function and the worst-case loss within a neighborhood of the model's parameters. SAM already has proven to be versatile across a diverse array of tasks such as meta-learning (Abbas et al., 2022), federated learning (Qu et al., 2022), vision models (Chen et al., 2021), or language models (Bahri et al., 2022).

Another desired property of the model is robustness, which could be improved when encouraging the model's parameters to satisfy strict conditions, i.e., SPD constraints (Gao et al., 2020), orthogonality, and full rank (Xie et al., 2017; Roy et al., 2019; Wang et al., 2020), etc. In those cases, the model's parameters are restricted to certain Riemannian manifolds, such as Grassmann, SPD, etc. Consequently, it becomes more challenging to work with its loss landscape, thus requiring novel optimization techniques that take into account the intrinsic geometry of the parameter spaces (Bonnabel, 2013; Luenberger, 1972; Kasai et al., 2019; Sato et al., 2019; Zhang et al., 2017).

In this work, we address both problems by bridging the gap between the sharpness-aware and Jacobian-aware optimizers on Riemannian manifolds. In doing so, we first derive a comprehensive theoretical analysis showing that the general loss function can be bounded by the empirical loss, the Riemannian gradient, and the projected Hessian. Motivated by this analysis, we proposed a Riemannian optimization technique named Riemannian Jacobian Regularization (RJR), which explicitly regularizes the Riemannian gradient norm and the Jacobian. Our empirical study shows that the RJR improves the model's generalization ability across a range of different tasks, namely supervised learning and self-supervised learning, for a diverse array of computer vision datasets (CIFAR100, CIFAR10, STL10, FGVCAircraft (Maji et al., 2013)), as well as different model's architectures (ResNet34, ResNet50, EfficientNetV2-S, EfficientNetV2-L Tan & Le (2021), and PyramidNet-101 (Han et al., 2017)). RJR has made a notable improvement upon SAM, SupCon

(Khosla et al., 2021), as well as other Riemannian optimizers, including Riemannian Stochastic Gradient Descent (RSGD) (Bonnabel, 2013), and Riemannian-SAM (Yun & Yang, 2023). In the ablation studies, we will also show the efficacy of RJR in simultaneously minimizing the Riemannian gradient norm and the Hessian spectral norm, thus indicating a flat region with low sharpness. In short, our contributions are as follows:

❶ We introduce a theoretical analysis that expresses the relationship between the general loss and the empirical loss via the Riemannian gradient and the projected Hessian.

❷ Motivated by this theoretical analysis, we introduce RJR, which strengthens the Jacobian regularization techniques to Riemannian manifolds. Empirical experiments across various settings show that RJR outperforms current methods by notable margins.

## 2 RELATED WORKS

**Optimization on Riemannian Manifolds.** Imposing appropriate constraints on model parameters has been shown to obtain the desired effect on the model performance (Roy et al., 2019; Absil et al., 2008a). In those situations, studying the intrinsic geometry of the parameters manifold could lead to improved optimization methods. For example, in the domain of metric learning, Roy et al. (2019) incorporated Stiefel manifolds to ensure that the learned parameters maintain orthogonality constraints. When the model is a Gaussian mixture, Gao et al. (2020) proposed a strategy involving learning on SPD manifolds to enforce SPD constraints. Furthermore, Grassmann manifolds have been utilized in encompassing recommender systems (Dai et al., 2012; Boumal & Absil, 2015) or modeling affine subspaces within document-specific language models (Hall & Hofmann, 2000). Since the optimization is carried out on manifolds, the Riemannian gradient descent approach developed by (Luenberger, 1972) is a tool to move on the manifold to look for minimums. Its stochastic version introduced by Bonnabel (2013) reduces computational overhead, thus gaining widespread adoption.

**Sharpness Aware Minimization and Jacobian Regularization.** The Sharpness-Aware Minimization (SAM) technique ( Foret et al. (2021a)) has gained prominence due to its effectiveness and scalability compared to previous methods. SAM's versatility is evident across various tasks and domains, making it a powerful optimization approach. SAM has found applications in diverse areas such as meta-learning bi-level optimization (Abbas et al., 2022), federated learning (Qu et al., 2022), vision models (Chen et al., 2021), language models (Bahri et al., 2022), domain generalization (Cha et al., 2021), and multi-task learning (Phan et al., 2022).

Recent works have further enhanced SAM's capabilities by exploring its underlying geometry (Kwon et al., 2021; Kim et al., 2022a), minimizing surrogate gaps (Zhuang et al., 2022), and speeding up training time (Du et al., 2022; Liu et al., 2022). Additionally, Kaddour et al. (2022a) empirically studied SAM's sharpness compared to Stochastic Weights Average (SWA) (Izmailov et al., 2018). In contrast, Möllenhoff & Khan (2023) demonstrated that SAM is an optimal Bayesian relaxation of standard Bayesian inference with a normal posterior. Moreover, Nguyen et al. (2023b) developed the sharpness concept for Bayesian Neural Networks. Nguyen et al. (2023a) generalized SAM by leveraging optimal transport-based distributional robustness with sharpness-aware minimization. Recently, Yun & Yang (2023) proposed Riemannian-SAM, which extends SAM to Riemannian manifolds, and the technique has demonstrated its efficacy on a wide range of manifolds. Lee et al. (2023) has extended SAM techniques to show that minimizing the Jacobian norm can affect the sharpness and the model accuracy, thus proposing explicitly regularizing the Jacobian norm.

## 3 PRELIMINARIES

### 3.1 FORMULATIONS AND NOTATIONS

This section presents the problem formulations and notions used for our theory development. We consider a classification problem where the data distribution denoted by $\mathcal{D}$, consists of pairs of $(\mathbf{x}, \mathbf{y})$, in which $\mathbf{x} \in \mathbb{R}^k$ and $\mathbf{y}$ belongs to one of the classes in $[C] = \{1, 2, \ldots, C\}$. We aim to construct a $C-$class classifier that maps $\mathbf{x}$ to its true corresponding label $\mathbf{y}$. This classifier is modeled by a function $f_{\boldsymbol{\theta}} : X \to Y$, parameterized by hyperparameter $\theta$, will produce a logit

vector $\mathbf{z} = f_{\boldsymbol{\theta}}(\mathbf{x})$ which in turn is used to predict the target labels. The model trainer is given a specific training set $\mathcal{S} = \{(\mathbf{x}_1, \mathbf{y}_1), (\mathbf{x}_2, \mathbf{y}_2), \cdots, (\mathbf{x}_n, \mathbf{y}_n)\}$ is then sampled from $\mathcal{D}$, those are i.i.d samples. Given $(\mathbf{x}, \mathbf{y}) \sim \mathcal{D}$, we use the per-sample loss function $\ell(f_{\boldsymbol{\theta}}(\mathbf{x}), \mathbf{y})$ to quantify the loss suffered by the model $f_{\boldsymbol{\theta}}$ when predicting $(\mathbf{x}, \mathbf{y})$. The empirical loss in the training set $\mathcal{S}$ is $\mathcal{L}_{\mathcal{S}}(\boldsymbol{\theta}) = \frac{1}{n}\sum_{i=1}^{n} \ell(f_{\boldsymbol{\theta}}(\mathbf{x}_i), \mathbf{y}_i)$, while the general loss in the data/label distribution $\mathcal{D}$ is $\mathcal{L}_{\mathcal{D}}(\boldsymbol{\theta}) = \mathbb{E}_{(\mathbf{x}, \mathbf{y}) \sim \mathcal{D}}[\ell(f_{\boldsymbol{\theta}}(\mathbf{x}), \mathbf{y})]$. Throughout our paper, $\|\mathbf{A}\|_{\sigma}$ denotes the spectral norm of a matrix $\mathbf{A}$, while $\|\mathbf{v}\|$ denotes the Euclidean norm-2 of a vector $\mathbf{v}$.

## 3.2 BACKGROUND ON RIEMANNIAN GEOMETRY

In this work, we assume that some conditions are imposed on the models (e.g., orthogonality, full rank, or SPD constraints), making the model parameters $\boldsymbol{\theta}$ lying in a *low-dimensional manifold* $\mathcal{M} \subset \mathbb{R}^k$ embedded in the *ambient vector space* $\mathbb{R}^k$, where the dimension $d$ of $\mathcal{M}$ is much smaller than $k$. Given a $\boldsymbol{\theta} \in \mathcal{M}$, denote $\mathcal{T}_{\boldsymbol{\theta}}\mathcal{M}$ as the *tangent space* of $\mathcal{M}$ at $\boldsymbol{\theta}$. Conventionally, $\boldsymbol{\theta}$ is the origin of $\mathcal{T}_{\boldsymbol{\theta}}\mathcal{M}$. Thus, $\epsilon \in \mathcal{T}_{\boldsymbol{\theta}}\mathcal{M}$ specifies the offset from $\boldsymbol{\theta}$ in the ambient vector space $\mathbb{R}^k$. The *tangent bundle* of $\mathcal{M}$ is defined as the disjoint union of the tangent spaces of $\mathcal{M}$:

$$\mathcal{T}\mathcal{M} = \{(\boldsymbol{\theta}, \mathbf{v}) : \boldsymbol{\theta} \in \mathcal{M} \text{ and } \mathbf{v} \in \mathcal{T}_{\boldsymbol{\theta}}\mathcal{M}\}.$$

For each $\boldsymbol{\theta} \in \mathcal{M}$, $\mathcal{T}_{\boldsymbol{\theta}}\mathcal{M}$ is linear space, thus one can define an inner product $\langle \cdot, \cdot \rangle_{\boldsymbol{\theta}} : \mathcal{T}_{\boldsymbol{\theta}}\mathcal{M} \times \mathcal{T}_{\boldsymbol{\theta}}\mathcal{M} \to \mathbb{R}$. A metric on $\mathcal{M}$ is a choice of inner product $\langle \cdot, \cdot \rangle_{\boldsymbol{\theta}}$ for each $\boldsymbol{\theta} \in \mathcal{M}$. The metric $\langle \cdot, \cdot \rangle_{\boldsymbol{\theta}}$ is a *Riemannian metric* if this metric varies smoothly with $\boldsymbol{\theta}$, in the sense that for all smooth vector fields $V, W$ on $\mathcal{M}$, the function $\boldsymbol{\theta} \mapsto \langle V(\boldsymbol{\theta}), W(\boldsymbol{\theta}) \rangle_{\boldsymbol{\theta}}$ is smooth from $\mathcal{M}$ to $\mathbb{R}$. A manifold with a Riemannian metric is called a *Riemannian manifold*.

For a given pair of $(\boldsymbol{\theta}, \mathbf{v}) \in \mathcal{T}\mathcal{M}$, there are many trajectories $c$ on the manifold $\mathcal{M}$ starting from $\boldsymbol{\theta}$ and follow the direction of $\mathbf{v}$, which can be formulated as $c : [0, 1] \to \mathcal{M} : c(0) = \boldsymbol{\theta}, c'(0) = \mathbf{v}$. A *retraction* picks a particular curve for each possible $(\boldsymbol{\theta}, \mathbf{v}) \in \mathcal{T}\mathcal{M}$. In particular, it is a smooth map

$$\mathcal{R} : \mathcal{T}\mathcal{M} \to \mathcal{M},$$

such that each curve $c(t) = \mathcal{R}(\boldsymbol{\theta}, t\mathbf{v})$ satisfies $c(0) = \boldsymbol{\theta}, c'(0) = \mathbf{v}$ for $0 \le t \le 1$. For the sake of simplification, we use $\mathcal{R}_{\boldsymbol{\theta}}(\mathbf{v})$ instead of $\mathcal{R}(\boldsymbol{\theta}, \mathbf{v})$.

Next, we want to define the Riemannian gradient of a smooth map $f : \mathcal{M} \to \mathbb{R}$. We start with the case $f : \mathcal{M} \to \mathcal{M}'$ being a smooth map between two general manifolds. For any tangent vector $\mathbf{v} \in \mathcal{T}_{\boldsymbol{\theta}}\mathcal{M}$, there exists a smooth curve $c$ on $\mathcal{M}$ passing through $\boldsymbol{\theta}$ with velocity $\mathbf{v}$. Then, $t \mapsto f(c(t))$ itself defines a curve on $\mathcal{M}'$ passing through $f(\boldsymbol{\theta})$, thus passing through $f(\boldsymbol{\theta})$ with a certain velocity. By definition, this velocity is a tangent vector of $\mathcal{M}'$ at $f(\boldsymbol{\theta})$. We call this tangent vector the *differential* of $f$ at $\boldsymbol{\theta}$ along $\mathbf{v}$. Specifically, the differential of $f : \mathcal{M} \to \mathcal{M}'$ at the point $\boldsymbol{\theta} \in \mathcal{M}$ is the linear map $Df(\boldsymbol{\theta}) : \mathcal{T}_{\boldsymbol{\theta}}\mathcal{M} \to \mathcal{T}_{f(\boldsymbol{\theta})}\mathcal{M}'$ defined by $Df(\boldsymbol{\theta})[\mathbf{v}] = \frac{d}{dt}f(c(t))\big|_{t=0}$, where $c$ is a smooth curve on $\mathcal{M}$ passing through $\boldsymbol{\theta}$ at $t = 0$ with velocity $\mathbf{v}$. For the case $\mathcal{M}' = \mathbb{R}$, which means $f$ is a smooth, real map, the *Riemannian gradient* of $f$ is defined as the unique vector field $\text{grad}_{\boldsymbol{\theta}} f$ on tangent space $\mathcal{T}_{\boldsymbol{\theta}}\mathcal{M}$ satisfies:

$$\forall (\boldsymbol{\theta}, \mathbf{v}) \in \mathcal{T}\mathcal{M}; Df(\boldsymbol{\theta})[\mathbf{v}] = \langle \mathbf{v}, \text{grad}_{\boldsymbol{\theta}} f(\boldsymbol{\theta}) \rangle_{\boldsymbol{\theta}},$$

in a neighbourhood of $\boldsymbol{\theta}$ on $\mathcal{M}$.

Finally, the orthogonal projection $\mathcal{P}_{\boldsymbol{\theta}}$ onto $\mathcal{T}_{\boldsymbol{\theta}}\mathcal{M}$ is defined as:

$$\mathcal{P}_{\boldsymbol{\theta}} : \mathbb{R}^k \to \mathcal{T}_{\boldsymbol{\theta}}\mathcal{M} : \mathbf{w} \mapsto \mathcal{P}_{\boldsymbol{\theta}}(\mathbf{w}),$$

with $\langle \mathbf{w} - \mathcal{P}_{\boldsymbol{\theta}}(\mathbf{w}), \mathbf{v} \rangle_{\boldsymbol{\theta}} = 0$ for all $\mathbf{v} \in \mathcal{T}_{\boldsymbol{\theta}}\mathcal{M}$. Once an orthogonal basis is chosen for $\mathcal{T}_{\boldsymbol{\theta}}\mathcal{M}$, $\mathcal{P}_{\boldsymbol{\theta}}$ is represented as a (symmetric) matrix. Thus, for readability purposes, the notation $\mathcal{P}_{\boldsymbol{\theta}}\mathbf{w}$ as a matrix multiplication is used instead of $\mathcal{P}_{\boldsymbol{\theta}}(\mathbf{w})$ as a linear map.

# 4 RIEMANNIAN JACOBIAN REGULARIZATION

## 4.1 THEORETICAL DEVELOPMENT

This section presents a theoretical development for Riemannian Jacobian Regularization (RJR). Consider the minimization problem in which the parameter space is an embedded manifold in $\mathbb{R}^k$

$$\min_{\boldsymbol{\theta} \in \mathcal{M}} \mathcal{L}_{\mathcal{D}}(\boldsymbol{\theta}).$$

The condition $\boldsymbol{\theta} \in \mathcal{M}$ can be interpreted as a constraint imposed on the optimization problem, such as orthogonality, full-rank, Euclidean, etc. There are two challenges on this problem, namely, (a) only a finite sample $\mathcal{S}$ is available instead of $\mathcal{D}$, thus one can only work with the empirical loss $\mathcal{L}_{\mathcal{S}}(\boldsymbol{\theta})$ rather than the generalization loss; (b) the minimization problem is constrained on the manifold $\mathcal{M}$. The next result will show that with a high probability, the generalization loss is upper-bounded by the empirical loss and some quantities characterizing the behavior of the loss function on the manifold $\mathcal{M}$. The proof can be found in Appendix C

**Theorem 1.** *Assume that the parameter space $\mathcal{M}$ is bounded and the loss function is Lipschitz. Then, for any $\rho > 0$ and $\delta \in [0; 1]$, with a probability of $1 - \delta$ over training set $\mathcal{S}$ generated from a distribution $\mathcal{D}$, we have the following inequality on the manifold $\mathcal{M}$:*

$$\mathcal{L}_{\mathcal{D}}(\boldsymbol{\theta}) \leq \mathcal{L}_{\mathcal{S}}(\boldsymbol{\theta}) + M\rho \|\mathrm{grad}_{\boldsymbol{\theta}} \mathcal{L}_{\mathcal{S}}(\boldsymbol{\theta})\|_{\boldsymbol{\theta}} + \frac{\rho^2}{2} \|\mathcal{P}_{\boldsymbol{\theta}} \nabla_{\boldsymbol{\theta}}^2 \mathcal{L}_{\mathcal{S}}(\boldsymbol{\theta})\|_{\sigma} + \mathcal{O}\left( \rho + D\sqrt{\frac{d \log \frac{1}{\rho} + \log \frac{1}{\delta}}{2n}} \right),$$

*for $M, D$ being constants and $d = \dim \mathcal{M}$. Here, $\mathcal{P}_{\boldsymbol{\theta}} \nabla_{\boldsymbol{\theta}}^2 \mathcal{L}_{\mathcal{S}}(\boldsymbol{\theta})$ is formed by projecting the columns of the Euclidean Hessian matrix $\nabla_{\boldsymbol{\theta}}^2 \mathcal{L}_{\mathcal{S}}(\boldsymbol{\theta})$ to the tangent space.*

A key novelty of the generalization inequality presented in Theorem 1 is that, unlike SAM (Foret et al., 2021b) or FisherSAM (Kim et al., 2022b), which bound the general loss using the worst-case empirical loss, our theorem directly relates the general loss to the empirical loss on the right-hand side. This approach suggests that to reduce the gap between the general loss and the empirical loss—i.e., to mitigate overfitting—we need to minimize both the gradient norm $\|\mathrm{grad}_{\boldsymbol{\theta}} \mathcal{L}_{\mathcal{S}}(\boldsymbol{\theta})\|_{\boldsymbol{\theta}}$ and the projected Hessian norm $\|\mathcal{P}_{\boldsymbol{\theta}} \nabla_{\boldsymbol{\theta}}^2 \mathcal{L}_{\mathcal{S}}(\boldsymbol{\theta})\|_{\sigma}$. As we will demonstrate in subsequent sections, regularizing the gradient norm $\|\mathrm{grad}_{\boldsymbol{\theta}} \mathcal{L}_{\mathcal{S}}(\boldsymbol{\theta})\|_{\boldsymbol{\theta}}$ implicitly minimizes sharpness, leading to sharpness-aware techniques such as Riemannian-SAM. Furthermore, minimizing both terms is expected to further reduce sharpness, which we empirically validate in Section 7.1. This reduction in sharpness, corresponds to a smaller generalization gap, thereby improving the generalization inequality.

Another key feature of this theorem is that our result is the first generalization equality that generalizes from the Euclidean spaces to the setting of Riemannian manifolds. By restricting the analysis to the embedding manifold, the error term is reduced to $\mathcal{O}(\sqrt{d})$, which is typically smaller than $\mathcal{O}(\sqrt{k})$ established in previous works such as Foret et al. (2021b) and Kim et al. (2022b), where $k$ represents the dimensionality of the ambient space. Building on this result, we introduce the Riemannian Jacobian Regularization (RJR) in the following section. As we show in subsequent sections, RJR effectively identifies low-sharpness regions on the manifold and, therefore, enhances generalization performance over previous methods, including SAM and Riemannian-SAM.

### 4.2 PRACTICAL ALGORITHM

Motivated by Theorem 1, we introduce the RJR algorithm that aims to simultaneously minimize the Riemannian gradient $\mathrm{grad}_{\boldsymbol{\theta}} \mathcal{L}_{\mathcal{S}}(\boldsymbol{\theta})$ and the term $\|\mathcal{P}_{\boldsymbol{\theta}} \nabla^2 \mathcal{L}_{\mathcal{S}}(\boldsymbol{\theta})\|_{\sigma}$ in the Inequality 1. The Riemannian gradient can be efficiently computed from the Euclidean gradient as:

$$\mathrm{grad}_{\boldsymbol{\theta}} \mathcal{L}_{\mathcal{S}}(\boldsymbol{\theta}) = \mathcal{P}_{\boldsymbol{\theta}} \nabla_{\boldsymbol{\theta}} \mathcal{L}_{\mathcal{S}}(\boldsymbol{\theta}). \tag{1}$$

It is important to note that *directly* computing the term $\|\mathcal{P}_{\boldsymbol{\theta}} \nabla_{\boldsymbol{\theta}}^2 \mathcal{L}_{\mathcal{S}}(\boldsymbol{\theta})\|_{\sigma}$ is prohibitively expensive. To *implicitly* regularize this term without computing it explicitly, we rely on the following lemma, with its proof provided in Appendix B.

**Lemma 1.** *We have the following bound:*

$$\|\mathcal{P}_{\boldsymbol{\theta}} \nabla^2 \mathcal{L}_{\mathcal{S}}\|_{\sigma} \approx \frac{1}{n} \Big\| \sum_{i=1}^{n} \mathcal{P}_{\boldsymbol{\theta}} \nabla_{\boldsymbol{\theta}} \mathbf{z}_i^{\top} \nabla_{\mathbf{z}_i}^2 \ell \nabla_{\boldsymbol{\theta}} \mathbf{z}_i \Big\| \leq \frac{\sqrt{\mathbb{E}_{\epsilon}[\|\mathrm{grad}_{\boldsymbol{\theta}}(\mathbf{z}\epsilon)\|^2] \mathbb{E}_{\epsilon}[\|\nabla_{\boldsymbol{\theta}}(\mathbf{z}\epsilon)\|^2]}}{2}, \tag{2}$$

*where the summation is taken over the training data $\mathcal{S}$ with $\mathbf{z}_i = f_{\boldsymbol{\theta}}(\mathbf{x}_i)$, $\epsilon$ is uniformly drawn from a unit hypersphere (i.e., $\epsilon \sim U(\mathbb{S}^{C-1})$), and $\mathbf{z} = \mathbb{E}_{S}[f_{\boldsymbol{\theta}}(\mathbf{x})]$.*

In this lemma, the first approximation comes from the Gaussian-Newton approximation, whose proof can be found in Lee et al. (2023). Motivated by Eq. (1) and Eq. (2), we propose to simultaneously minimize the terms $\|\mathcal{P}_{\boldsymbol{\theta}} \nabla_{\boldsymbol{\theta}} \mathcal{L}_{\mathcal{S}}(\boldsymbol{\theta})\|$, $\|\mathrm{grad}_{\boldsymbol{\theta}}(\mathbf{z}\epsilon)\|$, and $\|\nabla_{\boldsymbol{\theta}}(\mathbf{z}\epsilon)\|$ along with the empirical loss $\mathcal{L}_{\mathcal{S}}(\boldsymbol{\theta})$

---

**Algorithm 1** Riemannian Jacobian Regularization (RJR)

---

**Input:** Manifold $\mathcal{M}$, training set $\mathcal{S} \doteq \cup_{i=1}^{n}\{(\mathbf{x}_i, \mathbf{y}_i)\}$. Loss function $\ell : \mathcal{W} \times \mathcal{X} \times \mathcal{Y} \mapsto \mathbb{R}^+$, batch size $b$, learning rate $\eta > 0$, ascent step sizes $\lambda_1, \lambda_2 > 0$.

Initialize $\boldsymbol{\theta}_0 \in \mathcal{M}, t = 0$

**repeat**

    Sample mini batch $\mathcal{B} = \{(\mathbf{x}_i, \mathbf{y}_i)\}_{i=1}^{b}$ and $\epsilon \sim U(\mathbb{S}^{C-1})$

    Compute the batch Riemannian gradients $\mathrm{grad}_{\boldsymbol{\theta}_t} \mathcal{L}_{\mathcal{B}}(\boldsymbol{\theta}_t)$ and $\mathrm{grad}_{\boldsymbol{\theta}_t}(\mathbf{z}\epsilon)$ using Eq. (1)

    Compute $\delta_t = \lambda_1 \frac{\mathrm{grad}_{\boldsymbol{\theta}_t} \mathcal{L}_{\mathcal{B}}(\boldsymbol{\theta}_t)}{\|\mathrm{grad}_{\boldsymbol{\theta}_t} \mathcal{L}_{\mathcal{B}}(\boldsymbol{\theta}_t)\|} + \lambda_2 \left( \frac{\mathrm{grad}_{\boldsymbol{\theta}_t}(\mathbf{z}\epsilon)}{\|\mathrm{grad}_{\boldsymbol{\theta}_t}(\mathbf{z}\epsilon)\|_2} + \frac{\nabla_{\boldsymbol{\theta}_t}(\mathbf{z}\epsilon)}{\left\|\nabla_{\boldsymbol{\theta}_t}(\mathbf{z}\epsilon)\right\|_2} \right)$

    *Ascend step:* Compute $\widehat{\boldsymbol{\theta}}_t = \mathcal{R}_{\boldsymbol{\theta}_t}(\mathcal{P}_{\boldsymbol{\theta}_t} \delta_t)$

    *Descend step:* $\boldsymbol{\theta}_{t+1} = \mathcal{R}_{\boldsymbol{\theta}_t}\big(-\eta \mathrm{grad}_{\boldsymbol{\theta}_t}\big(\mathcal{L}_{\mathcal{B}}(\widehat{\boldsymbol{\theta}}_t)\big)\big)$

**until** *converges*

---

by regularizing these terms. Consider regularizing the first term alone, the Taylor expansion on smooth manifolds establishes that:

$$\mathcal{L}_{\mathcal{S}}\left( \mathcal{R}_{\boldsymbol{\theta}}\left( \lambda_1 \frac{\mathrm{grad}_{\boldsymbol{\theta}} \mathcal{L}_{\mathcal{S}}(\boldsymbol{\theta})}{\|\mathrm{grad}_{\boldsymbol{\theta}} \mathcal{L}_{\mathcal{S}}(\boldsymbol{\theta})\|} \right) \right) \approx \mathcal{L}_{\mathcal{S}}(\boldsymbol{\theta}) + \lambda_1 \mathrm{grad}_{\boldsymbol{\theta}} \mathcal{L}_{\mathcal{S}}(\boldsymbol{\theta})^{\top} \frac{\mathrm{grad}_{\boldsymbol{\theta}} \mathcal{L}_{\mathcal{S}}(\boldsymbol{\theta})}{\|\mathrm{grad}_{\boldsymbol{\theta}} \mathcal{L}_{\mathcal{S}}(\boldsymbol{\theta})\|} \tag{3}$$

$$= \mathcal{L}_{\mathcal{S}}(\boldsymbol{\theta}) + \lambda_1 \|\mathrm{grad}_{\boldsymbol{\theta}} \mathcal{L}_{\mathcal{S}}(\boldsymbol{\theta})\|. \tag{4}$$

Explicitly regularizing the term $\|\mathrm{grad}_{\boldsymbol{\theta}} \mathcal{L}_{\mathcal{S}}(\boldsymbol{\theta}))\|$ itself requires taking the gradient of a Riemannian gradient, giving a second-order term, which can be expensive. Instead, we propose to implicitly regularize $\|\mathrm{grad}_{\boldsymbol{\theta}} \mathcal{L}_{\mathcal{S}}(\boldsymbol{\theta})\|$ by minimizing the LHS of Eq. (4), which leads to the two gradient steps like SAM (Foret et al., 2021b)

$$\widehat{\boldsymbol{\theta}}_t = \mathcal{R}_{\boldsymbol{\theta}_t}(\delta_t)) \qquad \text{where} \qquad \delta_t = \lambda_1 \frac{\mathrm{grad}_{\boldsymbol{\theta}_t} \mathcal{L}_{\mathcal{S}}(\boldsymbol{\theta}_t)}{\|\mathrm{grad}_{\boldsymbol{\theta}_t} \mathcal{L}_{\mathcal{S}}(\boldsymbol{\theta}_t)|},$$

$$\boldsymbol{\theta}_{t+1} = \mathcal{R}_{\boldsymbol{\theta}_t}(-\eta \mathrm{grad}_{\boldsymbol{\theta}_t} L_{\mathcal{S}}(\widehat{\boldsymbol{\theta}}_t)).$$

Recall that besides $\|\mathrm{grad}_{\boldsymbol{\theta}} \mathcal{L}_{\mathcal{S}}(\boldsymbol{\theta}))\|$, we also want to implicitly regularize $\|\mathrm{grad}_{\boldsymbol{\theta}}(\mathbf{z}\epsilon)\|$ and $\|\nabla_{\boldsymbol{\theta}}(\mathbf{z}\epsilon)\|$. The approach above generalizes, leading to the modified ascending step

$$\delta_t = \lambda_1 \frac{\mathrm{grad}_{\boldsymbol{\theta}_t} \mathcal{L}_{\mathcal{S}}(\boldsymbol{\theta}_t)}{\|\mathrm{grad}_{\boldsymbol{\theta}_t} \mathcal{L}_{\mathcal{S}}(\boldsymbol{\theta}_t)|} + \lambda_2 \left( \frac{\mathrm{grad}_{\boldsymbol{\theta}_t}(\mathbf{z}\epsilon)}{\|\mathrm{grad}_{\boldsymbol{\theta}_t}(\mathbf{z}\epsilon)\|} + \frac{\nabla_{\boldsymbol{\theta}_t}(\mathbf{z}\epsilon)}{|\nabla_{\boldsymbol{\theta}_t}(\mathbf{z}\epsilon)|} \right).$$

This modified ascending step leads to the two gradient steps procedure:

$$\boldsymbol{\theta}_{t+1} = \mathcal{R}_{\boldsymbol{\theta}_t}(-\eta \mathrm{grad}_{\boldsymbol{\theta}_t}(\mathcal{L}_{\mathcal{B}}(\widehat{\boldsymbol{\theta}}_t))),$$

$$\widehat{\boldsymbol{\theta}}_t = \mathcal{R}_{\boldsymbol{\theta}_t}(\mathcal{P}_{\boldsymbol{\theta}_t} \delta_t),$$

$$\delta_t = \lambda_1 \frac{\mathrm{grad}_{\boldsymbol{\theta}_t} \mathcal{L}_{\mathcal{B}}(\boldsymbol{\theta}_t)}{\|\mathrm{grad}_{\boldsymbol{\theta}_t} \mathcal{L}_{\mathcal{B}}(\boldsymbol{\theta}_t)\|} + \lambda_2 \left( \frac{\mathrm{grad}_{\boldsymbol{\theta}_t}(\mathbf{z}\epsilon)}{\|\mathrm{grad}_{\boldsymbol{\theta}_t}(\mathbf{z}\epsilon)\|} + \frac{\nabla_{\boldsymbol{\theta}_t}(\mathbf{z}\epsilon)}{\|\nabla_{\boldsymbol{\theta}_t}(\mathbf{z}\epsilon)\|} \right).$$

where $\mathcal{B}$ is the sampled mini-batch, $\mathbf{z} = \sum_{\mathbf{x} \in \mathcal{B}} f_{\boldsymbol{\theta}}(\mathbf{x})$ and $\epsilon \sim U(\mathbb{S}^{C-1})$, leading to Algorithm 1.

In this algorithm, the Riemannian gradient $\mathrm{grad}_{\boldsymbol{\theta}_t} \mathcal{L}_{\mathcal{B}}(\boldsymbol{\theta}_t)$ can be obtained from the Euclidean counterpart $\nabla_{\boldsymbol{\theta}} \mathcal{L}_{\mathcal{B}}(\boldsymbol{\theta})$ using Eq. (1). Similarly, $\mathrm{grad}_{\boldsymbol{\theta}_t}(\mathbf{z}\epsilon)$ can be efficiently computed from $\nabla_{\boldsymbol{\theta}_t}(\mathbf{z}\epsilon)$. Moreover, the gradients $\nabla_{\boldsymbol{\theta}_t}(\mathbf{z}\epsilon)$ and $\nabla_{\boldsymbol{\theta}} \mathcal{L}_{\mathcal{B}}(\boldsymbol{\theta})$ are related through the Jacobian matrix, allowing for their joint efficient computation. Consequently, all the additional terms can be computed efficiently with a single backward pass, giving RJR the same computational complexity as SAM and Riemannian-SAM. Notably, RJR is a generalization upon many prior works for specific choices of manifolds and hyperparameters. If we set $\lambda_2 = 0$ for a general Riemannian manifold, we have Riemannian-SAM Yun & Yang (2023). On the other hand, if we choose the manifold to be Euclidean and $\lambda_1 = 0$, we obtain EJR Lee et al. (2023); if we set $\lambda_2 = 0$, we obtain SAM Foret et al. (2021b).

# 5 APPLICATIONS TO SUPERVISED AND SELF-SUPERVISED LEARNING

**1**This section presents the applications of RJR for three settings: *supervised learning*, *labeled self-supervised learning*, and *unlabeled self-supervised learning*. In the subsequent sections, we will empirically demonstrate the efficacy of our algorithm in contrast with the baselines on these applications. Throughout this paper, we particularly focus on the Stiefel manifolds:

**Definition 1** (The Stiefel Manifolds). *The set of $n \times p$ matrices, for $p \leq n$, with orthogonal columns and Frobenius inner products forms a Riemannian manifold is called the Stiefel manifold $St(p, n)$*

$$St(p, n) \doteq \{\mathbf{X} \in \mathbb{R}^{n \times p} : \mathbf{X}^\top \mathbf{X} = \mathbf{I}_p\}.$$

Absil et al. (2008b) proposed multiple retractions for Stiefel manifolds. For the sake of computational complexity, we suggest using the retraction: $R_{\mathbf{X}}(\varepsilon) = \text{qf}(\mathbf{X} + \varepsilon)$ in which $\text{qf}(\mathbf{A})$ denote the $\mathbf{Q}$ factor of the QR-decomposition of a matrix $\mathbf{A}$. Accordingly, the projection on the Stiefel manifolds can also be derived as $\mathcal{P}_{\mathbf{X}}(\mathbf{v}) = \mathbf{v} - \mathbf{X}\text{Sym}(\mathbf{X}^\top \mathbf{v})$ in which $\text{Sym}(\mathbf{A}) = \frac{1}{2}(\mathbf{A} + \mathbf{A}^\top)$. In this paper, we demonstrate the performance of the Stiefel manifold in two applications: *imposing orthogonal convolutional filters* in CNN and *metric learning for self-supervised learning*.

## 5.1 METRIC LEARNING FOR SELF-SUPERVISED LEARNING

We consider two self-supervised settings, including *labeled self-supervised learning* with the Supervised Contrastive (SupCon) methodology proposed by Khosla et al. (2021) and *unlabeled self-supervised learning* with the SimCLR loss function (Chen et al., 2020). Our settings are as follows: For a set of $N$ randomly sampled sample/label pairs, $\{\mathbf{x}_k, \mathbf{y}_k\}_{k=1}^N$, the corresponding batch used for training consists of $2N$ pairs, $\{\tilde{\mathbf{x}}_l, \tilde{\mathbf{y}}_l\}_{l=1}^{2N}$, where $\tilde{\mathbf{x}}_{2k}$ and $\tilde{\mathbf{x}}_{2k-1}$ are random augmentations of $\mathbf{x}_k$, and $\tilde{\mathbf{y}}_{2k-1} = \tilde{\mathbf{y}}_{2k} = \mathbf{y}_k$. A set of $N$ samples is referred to as a "batch," and the set of $2N$ samples is a "multiview batch". Within a multiview batch, let $i \in I = \{1, \cdots, 2N\}$ be the index of an arbitrary augmented sample, and let $j(i)$ be the index of the other augmented sample originating from the same source sample. The architecture

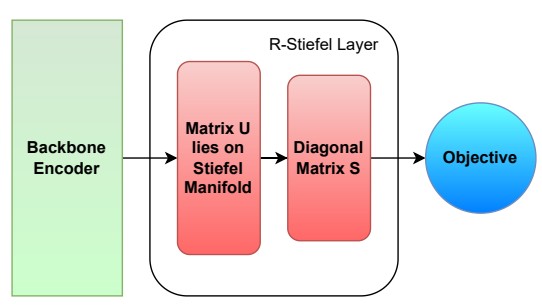

Figure 1: Metric learning with R-Stiefel layer. The linear projectional layer is replaced with the R-Stiefel layer consisting of $\mathbf{U} \in St(n, p)$ and a diagonal matrix $\mathbf{S}$.

of both settings involves two components: **1)** The backbone Encoders, which is denoted as $\text{Enc}(\cdot)$; and **2)** The projection head $P(\cdot)$, which is either a linear or fully-connected low-dimensional layer. It is worth noting that the projection head $P(\cdot)$ differs from the Riemannian projection operation $\mathcal{P}_{\boldsymbol{\theta}}$. For any $l$, denote $\mathbf{z}_l = P(\text{Enc}(\tilde{\mathbf{x}}_l))$.

As proposed by Khosla et al. (2021), the logit $\mathbf{z}_l$'s are then trained with the SupCon objective:

$$\mathcal{L}_{out}^{sup} := \sum_{i \in I} \mathcal{L}_{out,i}^{sup} = \sum_{i \in I} \frac{-1}{|C(i)|} \log \frac{\exp(\frac{\mathbf{z}_i \cdot \mathbf{z}_p}{\tau})}{\sum_{a \in A(i)} \exp(\frac{\mathbf{z}_i \cdot \mathbf{z}_a}{\tau})}$$

$$= \mathcal{L}(\mathbf{z}_1 \cdots, \mathbf{z}_{2N}) = \mathcal{L}(P(f(\tilde{\mathbf{x}}_1)) \cdots, P(f(\tilde{\mathbf{x}}_{2N}))),$$

where $A(i) = I \backslash \{i\}$, and $C(i) = \{c \in A(i) : \tilde{\mathbf{y}}_c = \tilde{\mathbf{y}}_i\}$.

On the other hand, SimCLR (Chen et al., 2020) defines the loss for a positive pair of examples as:

$$\ell_{i,j} = -\log \frac{\exp(s_{i,j}/\tau)}{\sum_{k=1}^{2N} \mathbb{1}_{k \neq i} \exp(s_{i,k}/\tau)},$$

where $s_{i,j} = \frac{\mathbf{z}_i \cdot \mathbf{z}_j}{\|\mathbf{z}_i\| \|\mathbf{z}_j\|}$ measures the similarity of the two logits, and $\mathbb{1}_{[k \neq i]}$ is an indicator function evaluating to 1 iff $k \neq i$. Then, the final loss is computed across all positive pairs in a mini-batch.

In our practical applications, the Euclidean inner product is replaced with the Mahalanobis distance in which $\langle \mathbf{h}_1, \mathbf{h}_2 \rangle = \mathbf{h}_1^\top \mathbf{M} \mathbf{h}_2$ with learnable $\mathbf{M}$. By doing so, $\mathbf{M}$ is learned to take into account

Table 1: Top-1 classification accuracy for supervised learning on Cross-Entropy loss function.

| | CIFAR10 | | | | | CIFAR100 | | | | | Aircraft | | | | | STL10 | | | | |
|---|---|---|---|---|---|---|---|---|---|---|---|---|---|---|---|---|---|---|---|---|
| Method | EffNetV2-S | EffNetV2-L | PyramidNet-101 | ResNet34 | ResNet50 | EffNetV2-S | EffNetV2-L | PyramidNet-101 | ResNet34 | ResNet50 | EffNetV2-S | EffNetV2-L | PyramidNet-101 | ResNet34 | ResNet50 | EffNetV2-S | EffNetV2-L | PyramidNet-101 | ResNet34 | ResNet50 |
| SGD | 89.7 | 91.5 | 94.2 | 94.8 | 94.5 | 67.0 | 68.7 | 77.6 | 73.6 | 74.6 | 80.8 | 81.0 | 82.3 | 78.7 | 82.4 | 73.0 | 73.1 | 77.6 | 70.9 | 69.9 |
| SAM | 90.2 | 92.1 | 95.9 | 95.5 | 95.3 | 70.2 | 70.3 | 79.1 | 75.0 | 75.0 | 81.0 | 82.3 | 83.1 | 80.5 | 82.7 | 75.2 | 78.2 | 80.0 | 73.0 | 76.0 |
| RSAM | 90.7 | 92.4 | 95.3 | 95.3 | 94.9 | 69.3 | 72.0 | 80.3 | 75.6 | 75.3 | 81.4 | 82.5 | 85.0 | 81.8 | 84.4 | 75.1 | 81.0 | 82.1 | 75.3 | 76.2 |
| RSGD | 90.0 | 92.2 | 94.9 | 95.8 | 95.7 | 68.0 | 70.8 | 79.8 | 74.9 | 77.1 | 81.9 | 82.1 | 83.2 | 83.5 | 84.0 | 74.1 | 79.6 | 81.6 | 76.0 | 76.1 |
| RJR | 91.7 | 94.2 | 96.3 | 96.4 | 96.5 | 73.6 | 76.4 | 83.3 | 77.3 | 78.6 | 84.5 | 85.0 | 89.0 | 84.6 | 85.2 | 79.9 | 81.2 | 84.0 | 78.3 | 79.3 |
| | (.31) | (.29) | (.25) | (.23) | (.19) | (.27) | (.24) | (.18) | (.25) | (.23) | (.31) | (.17) | (.32) | (.23) | (.35) | (.27) | (.19) | (.26) | (.31) | (.25) |

the local geometry of the parameter space, and the neighborhood becomes an adaptive ellipsoid instead of an open ball that treats every direction identically. Singular Value Decomposition yields $\mathbf{M} = \mathbf{U}\mathbf{D}\mathbf{U}^\top = \mathbf{U}\mathbf{D}^{1/2}\mathbf{D}^{1/2}\mathbf{U}^\top$. Denote $\mathbf{S} = \mathbf{D}^{1/2}$, it follows that:

$$\langle \mathbf{h}_1, \mathbf{h}_2 \rangle = \mathbf{h}_1^\top \mathbf{M} \mathbf{h}_2 = (\mathbf{h}_1^\top \mathbf{U}\mathbf{S}) \cdot (\mathbf{h}_2^\top \mathbf{U}\mathbf{S})^\top.$$

Motivated by the equation above, instead of optimizing $\mathcal{L}(P(\text{Enc}(\tilde{\mathbf{x}}_1)), \cdots, P(\text{Enc}(\tilde{\mathbf{x}}_{2N})))$, we will optimize $\mathcal{L}(P(\text{Enc}(\tilde{\mathbf{x}}_1))\mathbf{U}\mathbf{S}, \cdots, P(\text{Enc}(\tilde{\mathbf{x}}_{2N}))\mathbf{U}\mathbf{S})$ in which $\mathbf{U}$ is a rotational matrix on the Stiefel manifold, and $\mathbf{S}$ is a diagonal matrix. From now on, we will call the layer that multiplies with the matrix $\mathbf{U}\mathbf{S}$ an R-Stiefel layer, illustrated in Figure 1. Such modification can be done on the SupCon loss function and other different loss functions involving distance calculations such as triplet loss (Roy et al., 2019). Since $\mathbf{U}$ is enforced to lie on the Stiefel manifold, this orthogonal matrix will be optimized with RJR. Other parameters, including the backbone and the diagonal matrix $\mathbf{S}$, will be optimized by Euclidean optimizers such as SAM or SGD.

### 5.2 Orthogonal Convolutional Neural Network

In the literature of deep learning, enforcing orthogonality on the convolutional filters has established various significant benefits, such as alleviating gradient vanishing or exploding phenomenon (Xie et al., 2017), decorrelating the filter banks so that they learn distinct features (Wang et al., 2020), or stabilize the distribution of activations over layers within CNNs and make optimization more efficient (Rodríguez et al., 2016; Desjardins et al., 2015). Let $\{\mathbf{W}_i\}_{i=1}^D$ be the set of convolutional kernels for the $\ell$-th layer in which $\mathbf{W}_i \in \mathbb{R}^{WHM}$. To impose orthogonality, Previous works introduce orthogonal regularizers such as $\mathcal{L}_{\text{ortho}} = \frac{\lambda}{2} \sum_{i=1}^{D} \|\mathbf{W}_i^\top \mathbf{W}_i - \mathbf{I}\|_2^2$ (Xie et al., 2017), or a self-convolution regularization term of the kernels (Wang et al., 2020) to encourage orthogonality between the convolutional kernels. In this section, we propose eliminating those regularizers and strictly enforcing the kernels to be always orthogonal during training. To do so, we flatten the kernels $\mathbf{W}_i$ into the column vectors of shape $W \times H \times M$. Let $\mathbf{K}_\ell$ be the matrix with the columns formed by $\mathbf{W}_i's$. With RJR, we can enforce $\mathbf{K}_\ell$ to always lie on the Stiefel manifold $\text{St}(W \times H \times M, C)$ during training. Therefore, throughout training, $\mathbf{K}_\ell^\top \mathbf{K}_\ell = \mathbf{I}_d$ always holds, therefore guarantees orthonormality between the kernels on the layer $\ell$. The next section will demonstrate that imposing orthogonality onto a single convolutional layer in the middle of the architecture by training with RJR can notably improve generalization ability.

## 6 Experimental Results

To assess RJR's efficacy, we experimented with various vision datasets (including CIFAR10, CIFAR100, STL10, and Aircraft). We conducted three experiments: the standard supervised classification, labeled self-supervised learning, and unlabeled self-supervised learning. In all settings, we compare and contrast RJR with Momentum SGD, SAM, Riemannian-SAM (Yun & Yang, 2023), and Riemannian SGD (Bonnabel, 2013). All the experiments were trained for 500 epochs on Pytorch with

a Tesla V100 GPU with 40GB RAM. In all settings, the learning rate of RJR is set to 0.1 with a cosine annealing learning rate scheduler throughout the experiments. $\lambda_1$ in RJR is set to 0.5, and $\lambda_2$ is set to 0.01. All the models are trained with a batch size of 256 on CIFAR100, CIFAR10, and STL10 and a batch size of 64 on the Aircraft dataset for all methods. To measure the error, 10% of the training set was initially allocated as a validation set to tune the hyperparameters. After rigorous testing, we found $\lambda_1 = 0.5, \lambda_2 = 0.01$ to be robust default values, as reported in Table 4. Subsequently, we conducted five independent runs for each setting and report the mean accuracies along with the 95% confidence interval.

**Supervised Learning.** In this first setting, we examine the classification accuracy with a cross-entropy loss on five architectures, including ResNet34, ResNet50, PyramidNet-101, EfficientNetV2-S, and EfficientNetV2-L. In this setting, RJR is incorporated to force the orthogonality on the convolutional layers. Specifically, we imposed orthogonality on a single convolutional layer in the middle of the architecture in all settings. Table 1 shows that RJR generalizes better than the baselines in this standard training setting, with an improvement of 3% on average compared to Riemannian-SAM.

**Labeled Self-Supervised Learning.** In this second set of experiments, we compare RJR with the baselines on two architechtures including ResNet34 and ResNet50. This set of experiments has two stages. The SupCon objective is trained with the baseline methods in pretraining. Then, in the second stage, we conduct linear evaluation, that is, to freeze the parameters and train a linear classifier. We note that in the pre-trained step, the projectional layer of SGD and SAM are linear layers, while RJR's is the R-Stiefel layer as discussed in Section 5.1. Therefore, the applications of RJR in this setting are two-fold: RJR is used to impose orthogonality on the convolutional layers and used for the R-Stiefel during pretraining. As shown in Table 2, RJR consistently outperforms the baselines. Furthermore, we note that on ResNet50, RJR made a remarkable accuracy of 82.52% on CIFAR100, which outperforms 7% compared to SupCon with SGD on the same setting and consistently outperforms other baselines.

Table 2: Top-1 classification accuracy for *labeled self-supervised learning* settings with SupCon loss.

| Method | CIFAR100 | | CIFAR10 | | Aircraft | | STL10 | |
|---|---|---|---|---|---|---|---|---|
| | ResNet50 | ResNet34 | ResNet50 | ResNet34 | ResNet50 | ResNet34 | ResNet50 | ResNet34 |
| SGD | $75.29_{\pm.21}$ | $74.04_{\pm.23}$ | $95.99_{\pm.11}$ | $95.34_{\pm.14}$ | $82.03_{\pm.24}$ | $78.19_{\pm.32}$ | $83.33_{\pm.23}$ | $85.69_{\pm.19}$ |
| SAM | $76.73_{\pm.16}$ | $76.91_{\pm.15}$ | $96.31_{\pm.13}$ | $96.07_{\pm.22}$ | $82.84_{\pm.21}$ | $81.73_{\pm.13}$ | $85.02_{\pm.19}$ | $87.10_{\pm.24}$ |
| RSGD | $78.13_{\pm.17}$ | $77.32_{\pm.33}$ | $96.06_{\pm.19}$ | $\underline{96.25}_{\pm.09}$ | $83.38_{\pm.26}$ | $83.17_{\pm.27}$ | $84.23_{\pm.22}$ | $86.03_{\pm.24}$ |
| RSAM | $\underline{79.46}_{\pm.13}$ | $\underline{78.52}_{\pm.16}$ | $\underline{96.11}_{\pm.24}$ | $95.81_{\pm.22}$ | $\underline{84.02}_{\pm.13}$ | $\underline{84.37}_{\pm.21}$ | $\underline{88.35}_{\pm.19}$ | $\underline{87.21}_{\pm.18}$ |
| **RJR** | $\mathbf{82.52}_{\pm.22}$ | $\mathbf{81.12}_{\pm.22}$ | $\mathbf{96.74}_{\pm.20}$ | $\mathbf{96.81}_{\pm.19}$ | $\mathbf{89.93}_{\pm.31}$ | $\mathbf{87.52}_{\pm.25}$ | $\mathbf{91.04}_{\pm.23}$ | $\mathbf{90.14}_{\pm.29}$ |

**Unlabeled Self-Supervised Learning.** Similar to the previous set, this set of experiments has two stages. In the first stage, the model was trained with the SimCLR objective (Chen et al., 2020) instead. In this set of experiments, RJR also outperforms the baselines by a notable margin on average, especially on CIFAR100, where RJR outperforms conventional SimCLR with SGD by a margin of 7%. We refer to Table 3 for more details.

Table 3: Top-1 classification accuracy for *unlabeled self-supervised learning* with SimCLR loss

| Method | CIFAR100 | | CIFAR10 | | Aircraft | | STL10 | |
|---|---|---|---|---|---|---|---|---|
| | ResNet50 | ResNet34 | ResNet50 | ResNet34 | ResNet50 | ResNet34 | ResNet50 | ResNet34 |
| SGD | $65.65_{\pm.31}$ | $63.05_{\pm.31}$ | $92.98_{\pm.22}$ | $90.98_{\pm.17}$ | $61.20_{\pm.36}$ | $59.37_{\pm.38}$ | $65.35_{\pm.24}$ | $64.23_{\pm.31}$ |
| SAM | $67.24_{\pm.19}$ | $64.32_{\pm.29}$ | $93.11_{\pm.23}$ | $91.16_{\pm.19}$ | $63.17_{\pm.38}$ | $\underline{64.01}_{\pm.32}$ | $69.93_{\pm.26}$ | $67.83_{\pm.31}$ |
| RSGD | $66.31_{\pm.32}$ | $63.58_{\pm.20}$ | $93.50_{\pm.35}$ | $91.02_{\pm.31}$ | $\underline{65.01}_{\pm.21}$ | $63.93_{\pm.25}$ | $68.61_{\pm.41}$ | $67.02_{\pm.35}$ |
| RSAM | $\underline{69.12}_{\pm.35}$ | $\underline{67.26}_{\pm.33}$ | $\underline{94.27}_{\pm.33}$ | $\mathbf{92.14}_{\pm.31}$ | $64.97_{\pm.43}$ | $61.82_{\pm.34}$ | $\underline{70.39}_{\pm.21}$ | $\underline{70.08}_{\pm.29}$ |
| **RJR** | $\mathbf{72.71}_{\pm.23}$ | $\mathbf{71.04}_{\pm.27}$ | $\mathbf{95.82}_{\pm.27}$ | $\underline{92.03}_{\pm.24}$ | $\mathbf{67.79}_{\pm.41}$ | $\mathbf{65.25}_{\pm.35}$ | $\mathbf{73.03}_{\pm.34}$ | $\mathbf{73.32}_{\pm.32}$ |

As discussed in Section 4.2, RJR establishes the same *theoretical* complexity as SAM and Riemannian-SAM since all the additional terms can be computed efficiently in a single backward pass. However, RJR is expected to be slower due to additional computations involving the Jacobian and Riemannian gradients. Despite this, Appendix A.1 shows that the runtime difference is negligible, making

the trade-off worthwhile for the improved final performance. Besides, as demonstrated in Section 7.1, RJR effectively minimizes both the gradient norm and the Hessian spectral norm, in line with theoretical expectations. Furthermore, our algorithm identifies low-sharpness regions on the manifold, enhancing robustness. For a detailed behavioral comparison of SAM and RJR, we refer to Appendix A.2, which highlights the importance of manifold-based optimization and the effectiveness of our algorithm in minimizing both the loss function and sharpness.

# 7 ABLATION STUDIES

In this section, we perform several ablation studies to gain a deeper understanding of RJR's behavior, including its effectiveness in minimizing sharpness and its robustness to hyperparameter choices.

## 7.1 RJR VS. RIEMANNIAN-SAM: SHARPNESS AND HESSIAN SPECTRA

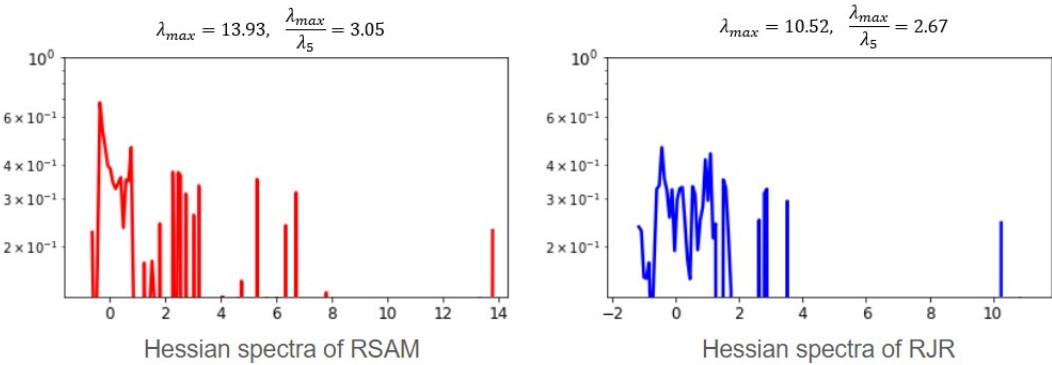

Figure 2: Hessian spectra of RSAM (left) vs. RJR (right). For RSAM, $\lambda_{\max} = 13.93$, $\frac{\lambda_{\max}}{\lambda_5} = 3.05$. For RJR, $\lambda_{\max} = 10.52$, $\frac{\lambda_{\max}}{\lambda_5} = 2.67$.

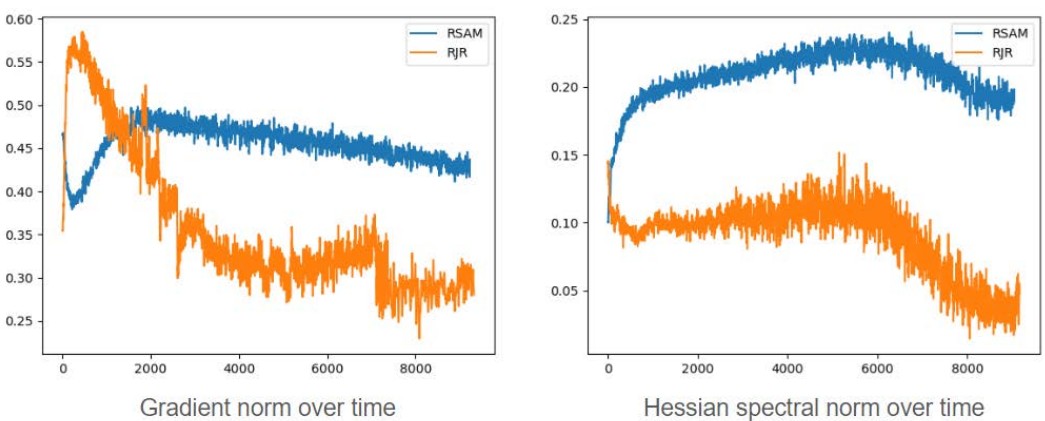

Figure 3: Gradient norms (left) and Hessian spectral norm (right) of Riemannian-SAM and RJR.

Throughout this work, we designed RJR to simultaneously minimize the empirical loss value, gradient norm, and Hessian spectral norm. To gain further insight into RJR's behavior and verify whether the algorithm successfully minimizes these three objectives, we first compare the Hessian spectrum of ResNet34 trained on CIFAR100 over 400 steps using RJR and Riemannian-SAM. As shown in Figure 2, the model trained with RJR exhibits a significantly lower maximum eigenvalue (10.52 for RJR compared to 13.93 for Riemannian-SAM) and a flatter eigenvalue distribution. We also assess the bulk

of the spectrum using the ratio $\lambda_{\max}/\lambda_5$, a commonly used proxy for sharpness (Jastrzebski et al., 2020), which yields values of 3.05 for Riemannian-SAM and 2.67 for RJR. Additionally, Figure 3 demonstrates that the gradient norm over time for RJR is notably lower than that of Riemannian-SAM, suggesting that RJR effectively minimizes both the Hessian spectral norm and the gradient norm, ultimately converging to minima with lower curvature on the manifold.

## 7.2 HYPERPARAMETERS SENSITIVITY

The implementation of RJR relies on two hyperparameters, $\lambda_1$ and $\lambda_2$. This ablation study investigates the performance of RJR on ResNet50 and PyramidNet-101 using the CIFAR100 dataset across various values for these hyperparameters. As shown in Tables 4 and 5, RJR demonstrated robust performance across a wide range of hyperparameter settings, indicating a desirable level of stability to these hyperparameters.

Table 4: Hyperparameter sensitivity on supervised setting with ResNet50 on CIFAR100 dataset.

| $\lambda_2$ \ $\lambda_1$ | 0.01 | 0.1 | 0.5 | 1 | 2 | 5 |
|---|---|---|---|---|---|---|
| 0 | 77.85 | 78.95 | 80.15 | 78.13 | 77.86 | 76.32 |
| 0.01 | 78.96 | 81.97 | **82.52** | 81.69 | 79.32 | 77.81 |
| 0.1 | 80.36 | 80.33 | 82.31 | 79.93 | 78.81 | < 70 |
| 1 | 77.51 | 78.09 | 77.58 | 78.37 | 74.11 | <70 |
| 2 | 71.31 | 73.11 | 72.09 | 73.11 | <70 | <70 |

Table 5: Hyperparameter sensitivity on supervised setting with PyramidNet-101 on CIFAR100 dataset.

| $\lambda_2$ \ $\lambda_1$ | 0 | 0.1 | 0.01 | 1 | 2 | 5 |
|---|---|---|---|---|---|---|
| 0 | 77.60 | 80.01 | 80.32 | 79.91 | 77.39 | 77.45 |
| 0.0001 | 81.02 | 81.89 | 83.35 | 82.06 | 78.50 | 76.87 |
| 1 | 79.81 | **83.47** | 80.79 | 80.08 | 78.65 | 76.01 |
| 2 | 78.30 | 81.44 | 81.94 | 78.35 | 77.36 | 75.23 |
| 5 | 77.14 | 77.10 | 78.15 | 77.46 | 76.71 | 78.09 |

## 8 CONCLUSION

We have extended the flat minimizers to differential manifolds by introducing a novel Riemannian optimizer. Theoretically, we presented a theorem that characterizes the generalization in terms of the Riemannian gradient and Hessian. Motivated by this analysis, we propose RJR that considers the intrinsic geometry and simultaneously minimizes the loss function, the Riemannian gradient norm, and the Jacobian. Empirically, RJR has demonstrated its effectiveness on different tasks with various datasets and consistently outperforms the comparative methods by a notable margin.

**Limitations and Future works.** Similar to other Riemannian optimizers such as Riemannian-SAM or RSGD, a limitation of RJR is that the method is only applicable to the class of Riemannian manifolds where the operations such as retraction and projection are well-defined. Even though this class of manifolds has demonstrated a wide range of applications in deep learning literature, further studies are still needed for a broader class of manifolds, such as general transformer architecture.

## REPRODUCIBILITY STATEMENT

Regarding the theoretical results, all the proof of the theories can be found in our appendix. Regarding the experiments, we have provided the necessary details to reproduce in Section 4.2 and Section 6, including the experimental settings, algorithm, hyperparameters details, and hardware details.

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

# A ADDITIONAL EXPERIMENTS

## A.1 WALL-CLOCK RUNTIME

In this ablation, we compare the single-epoch wall-clock runtimes of SGD, SAM, Riemannian-SAM, RSGD, and RJR. It is expected that SAM, Riemannian-SAM, and RJR take at least twice as long as SGD or RSGD because these methods involve double backward-forward computation in each iteration. Since the RJR requires additional computations such as the Riemannian gradients and the Jacobians as shown in Algorithm 1, it is expected that the RJR would take longer than Riemannian-SAM. However, as shown in Table 6, we emphasize that these additional computations can be done efficiently. In particular, while the RJR improves the classification accuracies by a notable margin, its wallclock runtime is only slightly slower than Riemannian-SAM by a 4% gap overall, therefore worth the tradeoff for better performance as well as robustness.

Table 6: Per-epoch wall-clock runtime in seconds.

| Method | CIFAR100 | | CIFAR10 | | AirCraft | |
|---|---|---|---|---|---|---|
| | RN34 | RN50 | RN34 | RN50 | RN34 | RN50 |
| SGD | $21.5_{\pm 1.73}$ | $40.1_{\pm 2.96}$ | $21.4_{\pm 1.72}$ | $38.8_{\pm 3.05}$ | $57.6_{\pm 2.59}$ | $114.5_{\pm 4.77}$ |
| RSGD | $30.5_{\pm 2.35}$ | $46.3_{\pm 3.87}$ | $28.1_{\pm 0.23}$ | $47.6_{\pm 1.52}$ | $64.8_{\pm 1.02}$ | $130.2_{\pm 3.25}$ |
| SAM | $49.1_{\pm 1.68}$ | $83.9_{\pm 2.79}$ | $48.8_{\pm 1.62}$ | $84.7_{\pm 2.68}$ | $125.3_{\pm 1.3}$ | $245.6_{\pm 4.30}$ |
| RSAM | $50.6_{\pm 1.66}$ | $85.7_{\pm 3.14}$ | $49.1_{\pm 1.90}$ | $88.3_{\pm 2.79}$ | $127.9_{\pm 2.1}$ | $253.2_{\pm 3.82}$ |
| RJR | $51.8_{\pm 2.97}$ | $88.6_{\pm 1.42}$ | $51.6_{\pm 1.03}$ | $92.4_{\pm 1.63}$ | $132.0_{\pm 1.4}$ | $260.8_{\pm 2.20}$ |

## A.2 RJR VS. SAM: BEHAVIORAL COMPARISON

In the previous section, we demonstrated the efficiency of RJR compared to Riemannian-SAM. In this section, we emphasize the importance of optimization on Riemannian manifolds in different scenarios and the effectiveness of RJR in accomplishing this task. We designed a simple experiment on the MNIST dataset to show a particular case where RJR is favorably robust compared to SAM, which did not take into account the intrinsic geometry. Indeed, we train a simple PCA-style autoencoder that aims to find an orthogonal matrix $\mathbf{W}$ which encodes each input $\mathbf{x}$ into lower-dimensional $\mathbf{z} = \mathbf{x}\mathbf{W}$, and then decodes as $\tilde{\mathbf{x}} = \mathbf{z}\mathbf{W}^{\top}$. The encoded vector $\mathbf{z}$ is then used for the classification. Therefore, the objective that we will minimize is the reconstruction loss, which is regularized with a classification loss. Since $\mathbf{W}$ is constrained to be orthogonal, we want it to stay within a Stiefel manifold during training. To enforce orthogonality with SAM, an *orthogonal regulazrizer* $\|\mathbf{W}^{\top}\mathbf{W} - \mathbf{I}_d\|_2^2$ is added, which gives the objective:

$$\mathcal{L}_{\mathcal{S}}(\mathbf{W}) = \frac{1}{n}\sum_{i=1}^{n}\|\mathbf{x}_i - \tilde{\mathbf{x}}_i\|_2^2 + \beta\mathrm{CrossEntropyLoss}(\mathbf{z}_i, \mathbf{y}_i) + \gamma\|\mathbf{W}^{\top}\mathbf{W} - \mathbf{I}_d\|_2^2.$$

To emphasize the importance of remaining in the Stiefel manifold during training in this case, we examine the effects attributable to the different values of $\gamma$ - the hyperparameter associated with the regularizer $\|\mathbf{W}^{\top}\mathbf{W} - \mathbf{I}_d\|_2^2$ that characterizes how orthogonal $\mathbf{W}$ is. In this set of experiments, we set the batch size to 16, the learning rate to 0.1, $\beta = 0.1$, and $\lambda_1 = 0.3, \lambda_2 = 0.01$. Figure 4 reports **1)** the loss value over time, **2)** the gradient norm of the loss function over time, and **3)** the values of the orthogonal regularizer, which measures how orthogonal the parameters were. For the convergence of the loss function, the smaller $\gamma$ is, the better SAM can keep up with RJR because the orthogonal regularization has less impact on the final loss function. However, in such cases, $\mathbf{W}$ fails to be orthogonal, demonstrating that SAM is remarkably sensitive to the orthogonal regularization. Hence, we emphasize that by explicitly enforcing $\mathbf{W}$ to be on the Stiefel manifold, RJR eliminates this vulnerability, remarkably reducing the sharpness and leading to better loss convergence as shown in Figure 4. In short, Figure 4 suggests that in certain scenarios, taking into account the intrinsic geometry of the parameters can notably enhance the model's robustness, and we claim that RJR is favorable to learn on manifolds in these scenarios.

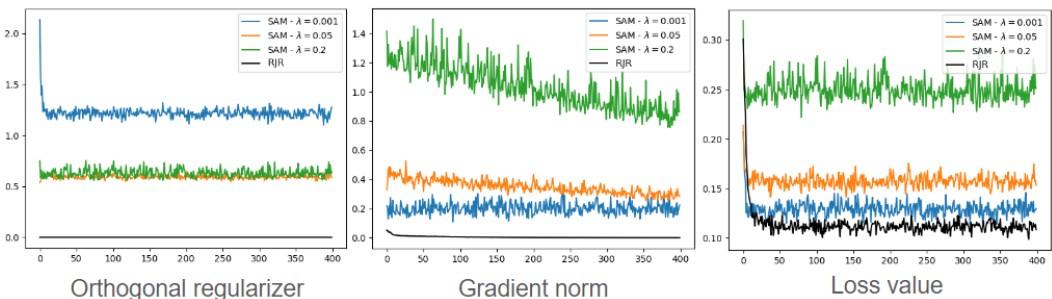

Figure 4: Comparision between SAM with different $\lambda$'s and RJR (black) on MNIST dataset regarding **a)** Loss value, **b)** Gradient norm, and **c)** Value of the orthogonal regularizer $\|\mathbf{W}^\top \mathbf{W} - \mathbf{I}_d\|_2^2$.

### A.3 MODEL ROBUSTNESS

In addition to its generalization ability, another desirable feature of the proposed approach is the robustness of the trained model. Recently, adversarial perturbations have been introduced as a way to assess the vulnerability of neural networks by considering the worst-case scenario under parameter corruption, which involves perturbation in the direction of the gradient (Sun et al., 2021). Specifically, the perturbation is defined as $\theta' = \theta + \alpha \frac{\nabla L_{\mathcal{S}}(\theta)}{\|\nabla L_{\mathcal{S}}(\theta)\|}$. It has been shown that sharpness-aware methods can improve model robustness (Kim et al., 2022b). In this section, we apply this perturbation to a ResNet50 model trained with RJR, Riemannian-SAM, and SGD on CIFAR10, with the perturbation strength $\alpha$ varying from $0$ to $4.0$. As shown in Figure 5, RJR exhibits less performance degradation as $\alpha$ increases compared to Riemannian-SAM and SGD.

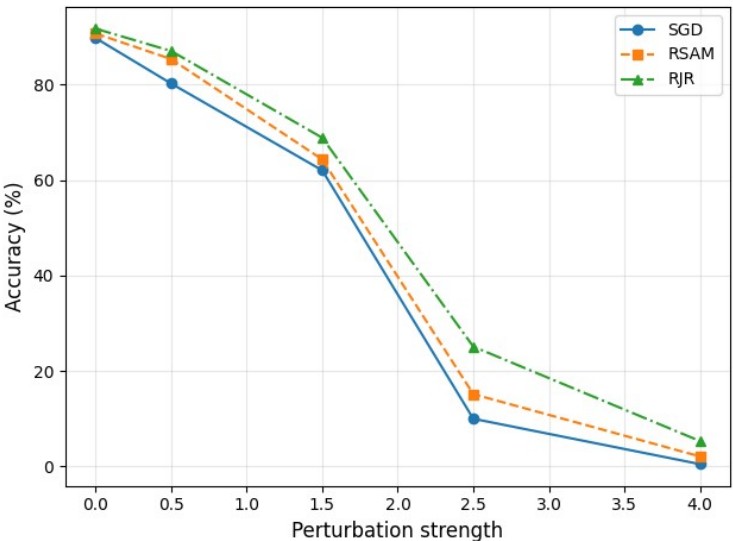

Figure 5: Robustness w.r.t adversarial parameter purturbation.

## B   PROOF OF LEMMA 1

*Proof.* In our proof, we introduce a few additional notations. Recall that the model $f_{\boldsymbol{\theta}}(\mathbf{x})$ outputs a logit vector $\mathbf{z} = f_{\boldsymbol{\theta}}(\mathbf{x})$. The logic vector $\mathbf{z}$ is then given as an input to the softmax function to yield a probability output $\mathbf{p} \equiv \text{Softmax}(\mathbf{z}) \in \Delta^{C-1}$ where $\Delta^{C-1} \equiv \{\mathbf{p} \in [0, 1]^C : \mathbf{1}^\top \mathbf{p} = 1, \mathbf{p} \geq 0\}$. We want the model to match the most probable class $\mathbf{c}_1$ to the true label $y$ where $\mathbf{c}(\mathbf{x}) \equiv \text{argsort}(\mathbf{p})$ in

descending order. We interchangeably denote the probability value corresponding to the true label $y$ as $p \equiv \mathbf{p}_y \in [0, 1]$. Notice that the logit Hessian matrix $\mathbf{M} \equiv \nabla_{\mathbf{z}}^2 \ell$ is fully characterized by $\mathbf{p}$, that is, $\mathbf{M} = \mathrm{diag}(\mathbf{p}) - \mathbf{p}\mathbf{p}^\top$. Since this is a rank-one modification of a diagonal matrix, we can obtain its eigenvalues $\{\lambda^{(i)}\}_{i=1}^C$ where $\lambda^{(i)}$ is the $i-$th largest eigenvalue of $\mathbf{M}$. We can also obtain the same ordered index of $(i) \in [C]$ with parentheses for the probability output $\mathbf{p} \in \Delta^{C-1}$, i.e. $\mathbf{c}_i = (i)$ and $\mathbf{p}_{(1)} \geq \mathbf{p}_{(2)} \geq \cdots \geq \mathbf{p}_{(C)}$, because the ordering is related to the eigenvalues $\{\lambda^{(i)}\}_{i=1}^C$ as shown in the following lemma, whose proof can be found in Theorem 4.1 by Lee et al. (2023)

**Lemma 2.** *(Eigen system of the logit Hessian $\mathbf{M}$) Then eigenvalues $\lambda^{(1)} \geq \lambda^{(2)} \geq \cdots \geq \lambda^{(C)}$ of the logit Hessian $\mathbf{M} \equiv \nabla_{\mathbf{z}}^2 \ell = \mathrm{diag}(\mathbf{p}) - \mathbf{p}\mathbf{p}^\top$ satisfies the following properties:*

1. $\mathbf{p}_{(i+1)} \leq \lambda^{(i)} \leq \mathbf{p}_{(i)}$ *for* $1 \leq i \leq C - 1$*, and* $\lambda^{(C)} = 0$

2. $\lambda^{(1)} \leq 2\mathbf{p}_{(1)}(1 - \mathbf{p}_{(1)})$

*Proof.* The proof can be found in Theorem 4.1 Lee et al. (2023). $\qquad \square$

Furthermore, we note that the Frobenius norm of the Jacobian can be efficiently computed with an unbiased estimator:

$$\|\mathbf{J}\|_F^2 = \mathbb{E}_{\epsilon \sim U(\mathbb{S}^{C-1})}[\|\mathbf{J}\epsilon\|^2] = \mathbb{E}_{\epsilon \sim U(\mathbb{S}^{C-1})}[\|\nabla_{\boldsymbol{\theta}}(\epsilon^\top \mathbf{z})\|^2].$$

Now we are ready to prove our lemma. Indeed, according to the Gauss-Newton approximation as derived by Lee et al. (2023), the Hessian of the loss function $\ell$ for a single sample $\mathbf{x}$ for the model parameter can be expressed as:

$$\nabla_{\boldsymbol{\theta}}^2 \ell = \nabla_{\boldsymbol{\theta}} \mathbf{z} \nabla_{\mathbf{z}}^2 \ell \nabla_{\boldsymbol{\theta}} \mathbf{z}^\top + \sum_{c=1}^C \nabla_{\boldsymbol{\theta}}^2 \mathbf{z}_c \nabla_{\mathbf{z}_c} \ell \approx \nabla_{\boldsymbol{\theta}} \mathbf{z} \nabla_{\mathbf{z}}^2 \ell \nabla_{\boldsymbol{\theta}} \mathbf{z}^\top.$$

For now, we denote $\mathbf{x}^{(i)}$ to be the $i-$th sample and denote $\mathbf{z}^{(i)}, \ell^{(i)}, \mathbf{J}^{(i)}$ to be the corresponding logit, the loss function, and the Jacobian for the given example $\mathbf{x}^{(i)}$, respectively. Notice that we have $\nabla_{\boldsymbol{\theta}} \mathbf{z} = \mathbf{J}$. Then, we have the following inequalities:

$$\|\mathcal{P}_{\boldsymbol{\theta}} \nabla^2 \mathcal{L}_{\mathcal{S}}\|_\sigma = \|\frac{1}{n} \sum_{i=1}^n \mathcal{P}_{\boldsymbol{\theta}} \nabla^2 \ell^{(i)}\|_\sigma \approx \|\frac{1}{n} \sum_{i=1}^n \mathcal{P}_{\boldsymbol{\theta}} \nabla_{\boldsymbol{\theta}}(\mathbf{z}^{(i)})^\top \nabla_{\mathbf{z}}^2 \ell^{(i)} \nabla_{\boldsymbol{\theta}} \mathbf{z}^{(i)}\|_\sigma$$

$$\leq \frac{1}{n} \sum_{i=1}^n \|\mathcal{P}_{\boldsymbol{\theta}} \nabla_{\boldsymbol{\theta}}(\mathbf{z}^{(i)})^\top \nabla_{\mathbf{z}}^2 \ell^{(i)} \nabla_{\boldsymbol{\theta}} \mathbf{z}^{(i)}\|_\sigma.$$

Utilizing the identities above along with Lemma 2, we have the following chain of inequalities:

$$\frac{1}{n} \sum_{i=1}^n \|\mathcal{P}_{\boldsymbol{\theta}} \nabla_{\boldsymbol{\theta}}(\mathbf{z}^{(i)})^\top \nabla_{\mathbf{z}}^2 \ell^{(i)} \nabla_{\boldsymbol{\theta}} \mathbf{z}^{(i)}\|_\sigma \leq \frac{1}{n} \sum_{i=1}^n \|\mathcal{P}_{\boldsymbol{\theta}} \mathbf{J}^{(i)}\|_\sigma \|\nabla_{\mathbf{z}}^2 \ell^{(i)}\|_\sigma \|\mathbf{J}^{(i)}\|_\sigma$$

$$\leq \frac{2\mathbf{p}_{(1)}(1 - \mathbf{p}_{(1)})}{n} \sum_{i=1}^n \|\mathcal{P}_{\boldsymbol{\theta}} \mathbf{J}^{(i)}\|_\sigma \|\mathbf{J}^{(i)}\|_\sigma$$

$$\leq \frac{2\mathbf{p}_{(1)}(1 - \mathbf{p}_{(1)})}{n} \sum_{i=1}^n \|\mathcal{P}_{\boldsymbol{\theta}} \mathbf{J}^{(i)}\|_F \|\mathbf{J}^{(i)}\|_F$$

$$\leq 2\mathbf{p}_{(1)}(1 - \mathbf{p}_{(1)}) \sqrt{\mathbb{E}_{\epsilon \sim U(\mathbb{S}^{C-1})}[\|\mathrm{grad}_{\boldsymbol{\theta}}(\mathbf{z}\epsilon)\|^2] \times \mathbb{E}_{\epsilon \sim U(\mathbb{S}^{C-1})}[\|\nabla_{\boldsymbol{\theta}}(\mathbf{z}\epsilon)\|^2]}$$

$$\leq \frac{\sqrt{\mathbb{E}_{\epsilon \sim U(\mathbb{S}^{C-1})}[\|\mathrm{grad}_{\boldsymbol{\theta}}(\mathbf{z}\epsilon)\|^2] \times \mathbb{E}_{\epsilon \sim U(\mathbb{S}^{C-1})}[\|\nabla_{\boldsymbol{\theta}}(\mathbf{z}\epsilon)\|^2]}}{2}$$

which concludes our proof. $\qquad \square$

To further proceed with our proof, we must introduce the concept of Riemannian Hessian. Indeed, the Riemannian Hessian of a function is defined as the covariant derivative of its gradient vector field for the Riemannian connection $\nabla$. More formally, we have the following definition:

**Definition 2.** *Let $\mathcal{M}$ be a Riemannian manifold with its Riemannian connection $\nabla$ (we refer to Boumal (2023) for more details about Riemannian connections). The Riemannian Hessian of $f$ at $x \in \mathcal{M}$ is the linear map $\mathrm{Hess}f(x) : \mathcal{T}_x\mathcal{M} \to \mathcal{T}_x\mathcal{M}$ defined as follows:*

$$\mathrm{Hess}f(x)[u] = \nabla_x \mathrm{grad}f.$$

We will make use of a property about Riemannian Hessian, whose proof can be found in Boumal (2023):

**Lemma 3.** *Let $\mathcal{M}$ be a Riemannian submanifold of an Euclidean space. Consider a smooth function $f : \mathcal{M} \to \mathbb{R}$. Let $\overline{G}$ be a smooth extension of $\mathrm{grad}f$, that is, $\overline{G}$ is any smooth vector field defined on a neighborhood of $\mathcal{M}$ in the embedding space such that $\overline{G}(x) = \mathrm{grad}f(x)$ for all $x \in \mathcal{M}$. Then, we have the following properties:*

$$\mathrm{Hess}f(x)[u] = \mathcal{P}_x(D\overline{G}(x)[u]).$$

We prove the following inequality regarding the Riemannian Hessian.

**Lemma 4.** *(Riemannian Hessian approximation) Suppose that $\boldsymbol{\theta} \in \mathcal{M}$, and $\epsilon \in \mathcal{T}_{\boldsymbol{\theta}}\mathcal{M}$. Then, we have the following inequality for some constant $m > 0$*

$$\langle \mathrm{Hess}\mathcal{L}_{\mathcal{S}}(\boldsymbol{\theta})[\epsilon], \epsilon \rangle_{\boldsymbol{\theta}} \le \|\epsilon\|^2 \left( m\|\mathrm{grad}_{\boldsymbol{\theta}}\mathcal{L}_{\mathcal{S}}(\boldsymbol{\theta})\| + \|\mathcal{P}_{\boldsymbol{\theta}}\nabla_{\boldsymbol{\theta}}^2\mathcal{L}_{\mathcal{S}}(\boldsymbol{\theta})\|_{\sigma} \right).$$

*Proof.* We start with the definition of the Hessian of a loss function:

$$\begin{aligned}
\mathrm{Hess}\mathcal{L}_{\mathcal{S}}(\boldsymbol{\theta})[\epsilon] &= \mathcal{P}_{\boldsymbol{\theta}}(D\mathrm{grad}_{\boldsymbol{\theta}}\mathcal{L}_{\mathcal{S}}(\boldsymbol{\theta})[\epsilon]) && \text{Def. of Riemannian Hessian} \\
&= \mathcal{P}_{\boldsymbol{\theta}}(D\mathcal{P}_{\boldsymbol{\theta}}\nabla_{\boldsymbol{\theta}}\mathcal{L}_{\mathcal{S}}(\boldsymbol{\theta})[\epsilon]) && \text{Property of Riemannian gradient} \\
&= \mathcal{P}_{\boldsymbol{\theta}} \langle \mathrm{grad}_{\boldsymbol{\theta}}(\mathcal{P}_{\boldsymbol{\theta}}\nabla_{\boldsymbol{\theta}}\mathcal{L}_{\mathcal{S}}(\boldsymbol{\theta})), \epsilon \rangle_{\boldsymbol{\theta}} && \text{Property of differential} \\
&= \mathcal{P}_{\boldsymbol{\theta}} \langle \mathcal{P}_{\boldsymbol{\theta}}\nabla_{\boldsymbol{\theta}}(\mathcal{P}_{\boldsymbol{\theta}}\nabla_{\boldsymbol{\theta}}\mathcal{L}_{\mathcal{S}}(\boldsymbol{\theta})), \epsilon \rangle_{\boldsymbol{\theta}} && \text{Property of Riemannian gradient} \\
&= \mathcal{P}_{\boldsymbol{\theta}} \left\langle \mathcal{P}_{\boldsymbol{\theta}}\nabla_{\boldsymbol{\theta}}^2\mathcal{L}_{\mathcal{S}}(\boldsymbol{\theta}) + \left( \mathcal{P}_{\boldsymbol{\theta}}\nabla_{\boldsymbol{\theta}}\mathcal{L}_{\mathcal{S}}(\boldsymbol{\theta}) \right)^{\top} \nabla(\mathcal{P}_{\boldsymbol{\theta}}^{\top}), \epsilon \right\rangle_{\boldsymbol{\theta}} && \text{Product rule} \\
&= \mathcal{P}_{\boldsymbol{\theta}} \left( \nabla_{\boldsymbol{\theta}}^2\mathcal{L}_{\mathcal{S}}(\boldsymbol{\theta}) \right)^{\top} \epsilon + \left( \mathcal{P}_{\boldsymbol{\theta}}\nabla_{\boldsymbol{\theta}}\mathcal{L}_{\mathcal{S}}(\boldsymbol{\theta}) \right)^{\top} \nabla(\mathcal{P}_{\boldsymbol{\theta}}^{\top}) \cdot \epsilon && \text{Def. of the Riemannian metric} \\
&= \mathcal{P}_{\boldsymbol{\theta}} \left( \nabla_{\boldsymbol{\theta}}^2\mathcal{L}_{\mathcal{S}}(\boldsymbol{\theta}) \right)^{\top} \epsilon + \left( \mathrm{grad}_{\boldsymbol{\theta}}\mathcal{L}_{\mathcal{S}}(\boldsymbol{\theta}) \right)^{\top} \nabla(\mathcal{P}_{\boldsymbol{\theta}}^{\top}) \cdot \epsilon && \text{Property of Riemannian gradient}
\end{aligned}$$

So, we have the following:

$$\langle \mathrm{Hess}\mathcal{L}_{\mathcal{S}}(\boldsymbol{\theta})[\epsilon], \epsilon \rangle_{\boldsymbol{\theta}} = \epsilon^{\top} \mathcal{P}_{\boldsymbol{\theta}}\nabla_{\boldsymbol{\theta}}^2\mathcal{L}_{\mathcal{S}}(\boldsymbol{\theta})\epsilon + \epsilon^{\top}(\mathrm{grad}_{\boldsymbol{\theta}}\mathcal{L}_{\mathcal{S}}(\boldsymbol{\theta}))^{\top}\nabla(\mathcal{P}_{\boldsymbol{\theta}}^{\top})\epsilon$$

Regarding the first term, we have the following inequality:

$$\epsilon^{\top}\mathcal{P}_{\boldsymbol{\theta}}\nabla_{\boldsymbol{\theta}}^2\mathcal{L}_{\mathcal{S}}(\boldsymbol{\theta})\epsilon \le \|\epsilon^{\top}\mathcal{P}_{\boldsymbol{\theta}}\nabla_{\boldsymbol{\theta}}^2\mathcal{L}_{\mathcal{S}}(\boldsymbol{\theta})\epsilon\| \le \|\mathcal{P}_{\boldsymbol{\theta}}\nabla_{\boldsymbol{\theta}}^2\mathcal{L}_{\mathcal{S}}(\boldsymbol{\theta})\|_{\sigma}\|\epsilon\|^2$$

Now we bound the second term. We have the following:

$$\|\epsilon^{\top}(\mathrm{grad}_{\boldsymbol{\theta}}\mathcal{L}_{\mathcal{S}}(\boldsymbol{\theta}))^{\top}\nabla(\mathcal{P}_{\boldsymbol{\theta}}^{\top})\epsilon\| \le \sum_{i=1}^{k}\sum_{j=1}^{k} \|\epsilon_i\epsilon_j\|\|\mathrm{grad}_{\boldsymbol{\theta}}\mathcal{L}_{\mathcal{S}}(\boldsymbol{\theta})^{\top}\nabla(\mathcal{P}_{\boldsymbol{\theta}}^{\top})_i\|$$

where we note that $\nabla(\mathcal{P}_{\boldsymbol{\theta}}^{\top})_i$ is the $i-$th column of a 3D tensor, and $\epsilon_i$ denotes the $-i$th element of $\epsilon$. Equivalently, it is the gradient of the $i-$th column of $\mathcal{P}_{\boldsymbol{\theta}}^{\top}$, that is $\nabla(\mathcal{P}_{\boldsymbol{\theta}}^{\top})_i$. Notice that $(\mathcal{P}_{\boldsymbol{\theta}}^{\top})_i$ is a unit vector. Also, $\mathcal{M}$ is assumed to be bounded and smooth, so there exists a constant $m = \mathcal{O}(k)$ such that $\|\nabla(\mathcal{P}_{\boldsymbol{\theta}}^{\top})_i\| \le \frac{m}{3}$ for all $i$. Hence, we have:

$$\sum_{i=1}^{k}\sum_{j=1}^{k} \|\epsilon_i\epsilon_j\|\|\mathrm{grad}_{\boldsymbol{\theta}}\mathcal{L}_{\mathcal{S}}(\boldsymbol{\theta})^{\top}\nabla(\mathcal{P}_{\boldsymbol{\theta}}^{\top})_i\| \le \frac{m}{3}\sum_{i=1}^{k}\sum_{j=1}^{k} \|\epsilon_i\epsilon_j\|\|\mathrm{grad}_{\boldsymbol{\theta}}\mathcal{L}_{\mathcal{S}}(\boldsymbol{\theta})\|$$

$$\le 3\frac{m}{3}\sum_{i=1}^{k} \|\epsilon_i\|^2\|\mathrm{grad}_{\boldsymbol{\theta}}\mathcal{L}_{\mathcal{S}}(\boldsymbol{\theta})\|$$

$$= m\|\mathrm{grad}_{\boldsymbol{\theta}}\mathcal{L}_{\mathcal{S}}(\boldsymbol{\theta})\|\|\epsilon\|^2.$$

Combining the two inequalities for two terms, we conclude the following inequality:

$$\langle \mathrm{Hess}\mathcal{L}_{\mathcal{S}}(\boldsymbol{\theta})[\epsilon], \epsilon \rangle_{\boldsymbol{\theta}} \le \|\epsilon\|^2 \left( m\|\mathrm{grad}_{\boldsymbol{\theta}}\mathcal{L}_{\mathcal{S}}(\boldsymbol{\theta})\| + \|\mathcal{P}_{\boldsymbol{\theta}}\nabla_{\boldsymbol{\theta}}^2\mathcal{L}_{\mathcal{S}}(\boldsymbol{\theta})\|_{\sigma} \right). \tag{5}$$

$\square$

## C  PROOF OF THEOREM 1

We first state several notations and settings that will be used throughout the proof. We are given a training dataset $\mathcal{S} = \{(\mathbf{x}_i, \mathbf{y}_i)\}_{i=1}^n$ drawn i.i.d from a distribution $\mathcal{D}$, we consider a family of models parameterized by $\boldsymbol{\theta} \in \mathcal{M} \subseteq \mathbb{R}^k$. Given a per-data-point loss function $\ell$, we recall the definition of the training set loss

$$\mathcal{L}_{\mathcal{S}}(\boldsymbol{\theta}) = \frac{1}{n} \sum_{i=1}^n \ell(\boldsymbol{\theta}, \mathbf{x}_i, \mathbf{y}_i)$$

and the population loss $\mathcal{L}_{\mathcal{D}}(\boldsymbol{\theta}) = \mathbb{E}_{(x,y)\sim\mathcal{D}}\big[\ell(\boldsymbol{\theta}, \mathbf{x}, \mathbf{y})\big]$. Suppose that the model space is an embedded Riemannian submanifold $\mathcal{M} \subset \mathbb{R}^k$ that has $d < k$ dimensions. Consider a point $\boldsymbol{\theta} \in \mathcal{M}$, denote by $\mathcal{T}_{\boldsymbol{\theta}}\mathcal{M}$ the tangent space of $\mathcal{M}$ at a point $\boldsymbol{\theta} \in \mathcal{M}$, which is homeomorphic to $\mathcal{M}$ and also has dimensionality of $d$.

Since $\mathcal{M}$ is assumed to be bounded, for every $\varepsilon > 0$, there exists a set $\{\boldsymbol{\theta}_i\}_{i=1}^J$ of predefined points on the manifold $\mathcal{M}$ that forms an $\varepsilon$-net of $\mathcal{M}$ with respect to the geodesic distance on $\mathcal{M}$. Indeed, for each $\boldsymbol{\theta} \in \mathcal{M}$, there exists $i$ such that $\boldsymbol{\theta}$ lies inside a neighborhood of $\boldsymbol{\theta}_i$ such that $d_{\mathcal{M}}(\boldsymbol{\theta}_i, \boldsymbol{\theta}) = d_i < \varepsilon$, in which $d_{\mathcal{M}}$ is the geodesic distance on manifold $\mathcal{M}$. So, we can define the following $J$ neighborhoods on $\mathcal{M}$ centered at $\boldsymbol{\theta}_j$:

$$R_j = \big\{ \boldsymbol{\theta} \in \mathcal{M} | d_{\mathcal{M}}(\boldsymbol{\theta}, \boldsymbol{\theta}_j) \le \varepsilon \big\}.$$

Now, we are ready to prove our main theorem. Indeed, we restate the theorem statement:

**Theorem 1.** *Assuming that the loss function $\mathcal{L}$ is $K-$Lipschitz and the parameter space is a bounded manifold. Then, for any small $\rho > 0$ and $\delta \in (0, 1)$, with a high probability $1 - \delta$ over the training set $\mathcal{S}$ generated from a distribution $\mathcal{D}$, the following holds:*

$$\mathcal{L}_{\mathcal{D}}(\boldsymbol{\theta}) \le \mathcal{L}_{\mathcal{S}}(\boldsymbol{\theta}) + M\rho\|\mathrm{grad}_{\boldsymbol{\theta}}\mathcal{L}_{\mathcal{S}}(\boldsymbol{\theta})\|_{\boldsymbol{\theta}} + \frac{\rho^2}{2}\|\mathcal{P}_{\boldsymbol{\theta}}\nabla^2_{\boldsymbol{\theta}}\mathcal{L}_{\mathcal{S}}(\boldsymbol{\theta})\|_{\sigma} + \mathcal{O}\left(\rho + D\sqrt{\frac{d\log\frac{1}{\rho} + \log\frac{1}{\delta}}{2n}}\right)$$

*for constants $M, D$ and $\mathcal{P}_{\boldsymbol{\theta}}\nabla^2\mathcal{L}_{\mathcal{S}}(\boldsymbol{\theta})$ is formed by projecting the columns of the Euclidean Hessian matrix $\nabla^2\mathcal{L}_{\mathcal{S}}(\boldsymbol{\theta})$ to the tangent space.*

*Proof.* Since $\mathcal{M}$ is assumed to be compact, it is bounded, and for every $\rho > 0$, there exists a set $\{\boldsymbol{\theta}_i\}_{i=1}^J$ of predefined points on the manifold $\mathcal{M}$ that forms an $\rho$-net of $\mathcal{M}$ with respect to the geodesic distance on $\mathcal{M}$. Indeed, for each $\boldsymbol{\theta} \in \mathcal{M}$, there exists $i$ such that $\boldsymbol{\theta}$ lies inside a neighborhood of $\boldsymbol{\theta}_i$ such that $d_{\mathcal{M}}(\boldsymbol{\theta}_i, \boldsymbol{\theta}) = d_i < \rho$, in which $d_{\mathcal{M}}$ is the geodesic distance on manifold $\mathcal{M}$. Since the loss function $\mathcal{L}$ is $K$-Lipschitz, we have:

$$\big|\mathcal{L}_{\mathcal{D}}(\boldsymbol{\theta}) - \mathcal{L}_{\mathcal{D}}(\boldsymbol{\theta}_i)\big| \le Kd_i = K\rho. \tag{6}$$

We have the following lemma regarding PAC-Bayes bound, whose proof can be found at Alquier (2023):

**Lemma 5.** *Suppose that $\boldsymbol{\theta} \in \boldsymbol{\theta}$, let $R(\boldsymbol{\theta}) = \mathbb{E}_{(X,Y)\sim P}[\ell(f_{\boldsymbol{\theta}}(X), Y)]$, and $r(\boldsymbol{\theta}) = \frac{1}{n}\sum_{i=1}^n \ell_i(\boldsymbol{\theta})$, assume that $\mathrm{card}(\boldsymbol{\theta}) = M < \infty$. Then, for any $\varepsilon \in (0, 1)$,*

$$\mathbb{P}_{\mathcal{S}}\left(\forall \boldsymbol{\theta} \in \boldsymbol{\theta}, R(\boldsymbol{\theta}) \le r(\boldsymbol{\theta}) + D\sqrt{\frac{\log\frac{M}{\varepsilon}}{2n}}\right) \ge 1 - \varepsilon$$

*Proof.* The proof can be found in Alquier (2023). □

Applying the lemma above for $\boldsymbol{\theta} = \{\boldsymbol{\theta}_j\}_{j=1}^J$, we have with a probability of at least $1 - \delta$, the following inequality holds:

$$\mathcal{L}_{\mathcal{D}}(\boldsymbol{\theta}_j) \le \mathcal{L}_{\mathcal{S}}(\boldsymbol{\theta}_j) + D\sqrt{\frac{\log\frac{J}{\delta}}{2n}}. \tag{7}$$

Let $\epsilon = \text{Log}_{\boldsymbol{\theta}}(\boldsymbol{\theta}_j)$ be the image of $\boldsymbol{\theta}_j$ under the logarithmic map, that is, the vector on $\mathcal{T}_{\boldsymbol{\theta}}\mathcal{M}$ such that $R_{\boldsymbol{\theta}}(\epsilon) = \boldsymbol{\theta}_j$. Notice that we have $d_{\mathcal{M}}(\boldsymbol{\theta}, \boldsymbol{\theta}_j) \leq \rho$, so $\|\epsilon\| \leq \rho$. Thus, with probability at least $1 - \delta$ we have the following inequalities

$$\mathcal{L}_{\mathcal{D}}(\boldsymbol{\theta}) \overset{6}{\leq} \mathcal{L}_{\mathcal{D}}(\boldsymbol{\theta}_j) + K\rho$$

$$\overset{7}{\leq} \mathcal{L}_{\mathcal{S}}(\boldsymbol{\theta}_j) + D\sqrt{\frac{\log \frac{J}{\delta}}{2n}} + K\rho$$

$$\leq \mathcal{L}_{\mathcal{S}}(R_{\boldsymbol{\theta}}(\epsilon)) + D\sqrt{\frac{\log \frac{J}{\delta}}{2n}} + 2K\rho$$

$$\leq \mathcal{L}_{\mathcal{S}}(\boldsymbol{\theta}) + \langle \text{grad}_{\boldsymbol{\theta}}\mathcal{L}_{\mathcal{S}}(\boldsymbol{\theta}), \epsilon \rangle_{\boldsymbol{\theta}} + \frac{1}{2}\langle \text{Hess}\mathcal{L}_{\mathcal{S}}(\boldsymbol{\theta})[\epsilon], \epsilon \rangle + \mathcal{O}(\|\epsilon\|^3) + D\sqrt{\frac{\log \frac{J}{\delta}}{2n}} + 2K\rho$$

$$\leq \mathcal{L}_{\mathcal{S}}(\boldsymbol{\theta}) + \|\text{grad}_{\boldsymbol{\theta}}\mathcal{L}_{\mathcal{S}}(\boldsymbol{\theta})\|_{\boldsymbol{\theta}}\|\epsilon\|_{\boldsymbol{\theta}} + \frac{1}{2}\epsilon^\top \text{Hess}\mathcal{L}_{\mathcal{S}}(\boldsymbol{\theta})[\epsilon] + 2K\rho + D\sqrt{\frac{\log \frac{J}{\delta}}{2n}}$$

$$\overset{5}{\leq} \mathcal{L}_{\mathcal{S}}(\boldsymbol{\theta}) + \|\text{grad}_{\boldsymbol{\theta}}\mathcal{L}_{\mathcal{S}}(\boldsymbol{\theta})\|_{\boldsymbol{\theta}}\|\epsilon\|_{\boldsymbol{\theta}} + \frac{1}{2}\|\epsilon\|^2\left(m\|\text{grad}_{\boldsymbol{\theta}}\mathcal{L}_{\mathcal{S}}(\boldsymbol{\theta})\| + \|\mathcal{P}_{\boldsymbol{\theta}}\nabla^2_{\boldsymbol{\theta}}\mathcal{L}_{\mathcal{S}}(\boldsymbol{\theta})\|_{\sigma}\right) + 2K\rho + D\sqrt{\frac{\log \frac{J}{\delta}}{2n}}$$

$$\leq \mathcal{L}_{\mathcal{S}}(\boldsymbol{\theta}) + \|\text{grad}_{\boldsymbol{\theta}}\mathcal{L}_{\mathcal{S}}(\boldsymbol{\theta})\|_{\boldsymbol{\theta}}\rho + \frac{1}{2}\rho^2\left(m\|\text{grad}_{\boldsymbol{\theta}}\mathcal{L}_{\mathcal{S}}(\boldsymbol{\theta})\| + \|\mathcal{P}_{\boldsymbol{\theta}}\nabla^2_{\boldsymbol{\theta}}\mathcal{L}_{\mathcal{S}}(\boldsymbol{\theta})\|_{\sigma}\right) + 2K\rho + D\sqrt{\frac{\log \frac{J}{\delta}}{2n}}.$$

When $\rho$ is small, we have $\rho^2(m\|\text{grad}_{\boldsymbol{\theta}}\mathcal{L}_{\mathcal{S}}(\boldsymbol{\theta})\|) \ll \rho\|\text{grad}_{\boldsymbol{\theta}}\mathcal{L}_{\mathcal{S}}(\boldsymbol{\theta})\|_{\boldsymbol{\theta}}$. Hence, there is a constant $M$ such that:

$$\mathcal{L}_{\mathcal{D}}(\boldsymbol{\theta}) \leq \mathcal{L}_{\mathcal{S}}(\boldsymbol{\theta}) + M\rho\|\text{grad}_{\boldsymbol{\theta}}\mathcal{L}_{\mathcal{S}}(\boldsymbol{\theta})\|_{\boldsymbol{\theta}} + \frac{\rho^2}{2}\|\mathcal{P}_{\boldsymbol{\theta}}\nabla^2_{\boldsymbol{\theta}}\mathcal{L}_{\mathcal{S}}(\boldsymbol{\theta})\|_{\sigma} + 2K\rho + D\sqrt{\frac{\log \frac{J}{\delta}}{2n}}$$

for some constant $M > 0$. Now we are left to bound $J$, the number of $\rho-$balls covering $\mathcal{M}$. Recall that $\mathcal{M}$ is a $d-$dimensional manifold covered within $J$ $\rho$-balls. If we denote $R_j$ to be the $\rho-$ball with the center $\boldsymbol{\theta}_i$, then $\text{vol}(R_j) = \mathcal{O}(\rho^d)$, implying $J = \mathcal{O}(\max_j \text{diam}(\mathcal{M})^d/\rho^d)$, thus $\log J = \mathcal{O}(d\log \frac{K}{\rho})$, in which $K = \text{diam}(\mathcal{M}) < +\infty$ since we assume $\mathcal{M}$ is bounded.

Hence, we conclude that:

$$\mathcal{L}_{\mathcal{D}}(\boldsymbol{\theta}) \leq \mathcal{L}_{\mathcal{S}}(\boldsymbol{\theta}) + M\rho\|\text{grad}_{\boldsymbol{\theta}}\mathcal{L}_{\mathcal{S}}(\boldsymbol{\theta})\|_{\boldsymbol{\theta}} + \frac{\rho^2}{2}\|\mathcal{P}_{\boldsymbol{\theta}}\nabla^2_{\boldsymbol{\theta}}\mathcal{L}_{\mathcal{S}}(\boldsymbol{\theta})\|_{\sigma} + \mathcal{O}\left(\rho + D\sqrt{\frac{d\log \frac{1}{\rho} + \log \frac{1}{\delta}}{2n}}\right).$$

$\square$

