# OpenReview forum: "Optimization on Manifolds with Riemannian Jacobian Regularization"
_ICLR.cc/2025/Conference — Submitted to ICLR 2025_

### Official Review · Reviewer_ynnr · 2024-10-30

**Soundness:** 2
**Presentation:** 2
**Contribution:** 2
**Rating:** 5
**Confidence:** 4

**Summary:**

In this work, a theoretical analysis that characterizes the relationship between the general loss and the perturbation of the empirical loss in the context of Riemannian manifolds is presented. Motivated by the result obtained from this analysis, the authors introduce an algorithm named Riemannian Jacobian Regularization (RJR), which explicitly regularizes the Riemannian gradient norm and the projected Hessian. Some experiments have been conducted to verify the performance of the proposed method.

**Strengths:**

1.	A theoretical analysis that expresses the relationship between the general loss and the empirical loss via the Riemannian gradient and the projected Hessian is provided.
2.	A Riemannian Jacobian Regularization (RJR) method is introduced to strengthen the Jacobian regularization techniques to Riemannian manifolds.

**Weaknesses:**

1.	The contributions of the work is not enough with only a theoretic analysis and a proposed method. I suggest the authors give more description about the proposed method, such as the role of each component played in the method.
2.	On page three, line 113, the definition of the Euclidean norm of a vector is not standardized.
3.	Why are some formulas not numbered, while others are?
4.	Only four compared methods are used, just from two references in 2013 and 2023. How to effectively validate the SOTA performance of this method? The following methods can be compared “Riemannian Manifold Learning, TPAMI, 2008”, “Generalized Learning Riemannian Space Quantization: A Case Study on Riemannian Manifold of SPD Matrices, TNNLS, 2021”, “Kernel Methods on Riemannian Manifolds with Gaussian RBF Kernels, TPAMI, 2015”.
5.	What is the Lemma 2 in the Appendix (A.2)? No related contents are described in the main contents of the paper. The given Lemma 2 in A. 2 is just already existed in Ref. Lee et al. (2023), as the authors described.
6.	How to obtain the first inequality on page 19, line 974? Similar as in the third inequality?

**Questions:**

Please see the weaknesses.

---

> ### Author Response · Authors · 2024-11-25
> **Response to Reviewer ynnr**
>
> We thank the reviewer for the constructive feedback and would like to address the concerns of the reviewer as follows:
>
> + **Regarding the theoretical contribution:** We appreciate the opportunity to elaborate on the theoretical contribution of our work. In the main theorem, we establish a generalization inequality for Riemannian manifolds, which, to the best of our knowledge, has not been previously reported, as prior results are limited to the Euclidean setting. A key distinction of our approach is that we directly bound the population loss using the empirical loss, rather than the worst-case empirical loss. To better understand the role of each component, as discussed in lines 230–237, if we focus solely on regularizing the gradient norm, we obtain a Riemannian sharpness-aware algorithm similar to Riemannian-SAM. However, our theorem motivates us to also regularize the Hessian term, which further reduces the gap between the general loss and the empirical loss, offering an improvement over Riemannian-SAM and other baselines. While Jacobian regularization has been previously introduced in the Euclidean context, it has not been theoretically justified. The Hessian term in our theorem justifies the use of Jacobian regularization as a means to further reduce sharpness, a result that is empirically supported by our ablation studies. Based on the reviewer's suggestions, we have incorporated this explanation in the rebuttal revision.
>
> + **Regarding the notation of the Euclidean norm:** We use the notation $\|\cdot\|$ instead of the standardized norm-2 notations $\|\cdot\|_2$ for shortness, in which we omitted the subscript for shortness. We have included a clarification in the rebuttal revision based on the reviewer's suggestion.
>
> + **Regarding the numbering of the Equations:** In our theoretical development, only a few equations are referred to later in the derivation. Then, for readability purposes, we decided to label only the equations that are being referred to.
>
> + **Regarding the first inequality on page 19, line 974:** In Equation (5), we have derived that
> $$|L_D(\theta)-L_D(\theta_i)|\leq K\rho$$
> Therefore, we have
> $$L_D(\theta)-L_D(\theta_i)\leq |L_D(\theta)-L_D(\theta_i)|\leq K\rho$$
> which gives the inequality at Line 974.
>
> + **Comparison with SOTA Methods:** Our work focuses on establishing a theoretical framework for a novel Riemannian optimizer. To illustrate the benefits of sharpness-aware Jacobian regularization on manifolds, we compare our algorithm with foundational optimizers, including SGD, SAM, and their Riemannian counterparts. We deliberately chose not to include comparisons with SOTA methods that diverge significantly from these fundamental optimizers in methodology, as this could detract from our primary theoretical focus. However, we greatly appreciate your suggestion. Due to time constraints, we will prioritize exploring them in future studies.
>
> + **Regarding the Appendix A.2:** The lemma that the appendix refers to is the Lemma 1 in the main text. According to the suggestion of the reviewer, we have fixed this issue in the rebuttal revision.

---

### Official Review · Reviewer_Rjrt · 2024-11-02

**Soundness:** 2
**Presentation:** 2
**Contribution:** 2
**Rating:** 3
**Confidence:** 4

**Summary:**

This paper introduces Riemannian Jacobian Regularization (RJR), a method that combines sharpness-aware and Jacobian regularization on Riemannian manifolds to improve generalization and robustness. The approach is tested on various supervised and self-supervised tasks.

**Strengths:**

RJR presents a straightforward optimization technique that improves generalization by explicitly controlling sharpness on Riemannian manifolds.

**Weaknesses:**

1. In Thm.1, why there is an ambient inner product, instead of the Riemannian ones? Will this cause the loss of some information?
2. Experiments are not convincing
   - Most datasets are small
   - The reason for not comparing it with RSAM under the same datasets as the original paper is unclear.
   - A common counterpart is the trivialization [1]. Trivializations could be faster and sometimes even better than the Riemannian ones. The missing comparison makes the empirical validation not convincing.
   - For the orthogonal constraints, why do not compare with some orthogonal tricks [2] is not clear
   - There are Riemannian networks, where data and parameters naturally lie in the manifold, such as SPD [3-4], Grassmannian [5-6], Lie groups [7], and hyperbolic [8]. How about the effects on these networks? Some works prefer to use trivialization, such as [6] and its previous work.
3. Some description lacks clarity:
   - some abbreviations come without their full name, such as SWA.
   - Typos: $\theta$ and $\mathcal{T}_\theta$​ in lines 120
   - L152: $\langle, \rangle$​ is the Euclidean one in the ambient space.
   - The readability of some proofs are poor. use \stackrel for each (key) derivation is a good habit (such as L974-992).


[1] Trivializations for Gradient-Based Optimization on Manifolds

[2] Cheap Orthogonal Constraints in Neural Networks: A Simple Parametrization of the Orthogonal and Unitary Group

[3] SPD domain-specific batch normalization to crack interpretable unsupervised domain adaptation in EEG

[4] ManifoldNet: A Deep Neural Network for Manifold-Valued Data With Applications

[5] Building Deep Networks on Grassmann Manifolds

[6] Matrix Manifold Neural Networks++

[7] A Lie Group Approach to Riemannian Batch Normalization

[8] Hyperbolic Neural Networks

**Questions:**

1. L 982-985: how to transform the Riemannian metric into the ambient inner product?
2. could we do  (Jacobian) regularization by trivialization? if so what is the advantage of Riemannian regularization?

---

> ### Author Response · Authors · 2024-11-25
> **Response to Reviewer Rjrt**
>
> We would like to thank the reviewer for the insightful feedback. We would like to address the concerns below:
>
> + **Regarding the inner product/norm in Theorem 1:** The first norm, corresponding to the Riemannian gradient term, should be expressed as the Riemannian norm. Similarly, the norm mentioned in lines 982–985 also refers to the Riemannian norm. We have corrected these typos in the revised manuscript and appreciate the reviewer for bringing this to our attention. Thank you for highlighting this oversight.
>
> + **Regarding the experiments on additional datasets:** In response to the reviewer's suggestion, we conducted additional experiments on three more challenging datasets: TinyImageNet, Caltech101, and CUB-200-2011. The results, obtained using ResNet34 in the supervised setting, are summarized below. These experiments further validate the effectiveness and robustness of our approach across diverse and complex datasets.
>
> | Methods | Tiny-Imagenet | Caltech101| CUB-200-2011 |
> |-------------|---------------|-----------|--------------|
> | SGD         |      57.65    |   69.19   |     52.97    |
> | SAM         |      60.27    |   70.38   |     54.72    |
> | R_SGD       |      59.34    |   69.02   |     53.31    |
> | R_SAM       |      61.55    |   71.46   |     56.22    |
> | RJR(Ours)   |      62.22    |   72.12   |     56.88    |
>
> The result shows that our method still outperforms the baselines.
>
> + **Regarding the choice of datasets to compare with Riemannian-SAM:** Since the source code for Riemannian-SAM was not released, we opted to conduct experiments under two different settings: supervised and unsupervised learning for image classification. To ensure the fairness and reliability of the comparisons, we maintained consistent experimental settings across all methods evaluated. This consistency eliminates potential biases and highlights the relative performance of each approach under identical conditions.
>
> + **Regarding the comparisons with other orthogonal tricks:** In Appendix A.1.2, we present an ablation study to highlight the superiority of our method compared to an alternative orthogonality technique proposed in [1]. Specifically, we designed a toy experiment to benchmark our approach against the method in [1], which regularizes with the term $|W^\top W - I_d|^2_2$ to encourage orthogonality in $W$. As illustrated in Figure 4, our method significantly outperforms this orthogonality trick in the toy example, demonstrating the advantages of learning on manifolds and the effectiveness of our approach. This result underscores the value of our method in achieving improved performance through principled manifold-based learning.
>
> + **Comparision with Trivialization [2]:** Firstly, we note that Jacobian Regularization can be applied alongside Trivialization. Specifically, this involves replacing the base optimizer with RJR, where the manifold of interest is simplified to Euclidean space. Following the reviewer’s suggestion, we conducted additional comparisons, incorporating Trivialization with both SAM and Jacobian Regularization as the base optimizers. The results of these experiments are summarized below.
>
> |Methods |Tiny-Imagenet|CIFAR100|STL10|
> |-|-|-|-|
> |RJR|60.93|76.43|81.33|
> |Trivialization + SAM|60.76|76.12|80.23|
> |Trivialization + Jacobian Regularization|62.81|77.03|82.07|
>
> The results indicate that Trivialization with Jacobian Regularization achieves a slight improvement over both methods. The key insight behind this improvement is that regularizing the Jacobian reduces the generalization gap more effectively than sharpness-aware methods, as supported by Theorem 1. Additionally, we observed that Trivialization has a notably faster runtime. Although time constraints prevented us from conducting a comprehensive evaluation of Trivialization, the reviewer’s comment highlights a promising avenue for future research. Specifically, developing a deeper theoretical analysis of Trivialization's generalization ability could provide valuable insights. We sincerely thank the reviewer for this thoughtful suggestion.
>
> + **Regarding other minor issues:** We have addressed the minor issues highlighted by the reviewer, including typos and abbreviations. These corrections have been incorporated into the revised rebuttal. We appreciate the reviewer’s attention to detail and thank them for pointing out these issues.
>
> **References:**
>
> [1] Jiayun Wang, Yubei Chen, Rudrasis Chakraborty, and Stella X. Yu. Orthogonal convolutional neural networks, 2020.
>
> [2] Lezcano-Casado, M. (2019). Trivializations for Gradient-Based Optimization on Manifolds. In Advances in Neural Information Processing Systems (pp. 9154–9164).

---

> ### Comment · Reviewer_Rjrt · 2024-11-25
>
> Thanks for the response.
> - The baseline seems weak, it hardly can tell if the benefits come from the network itself or the optimizer. Convincing experiments should be conducted on strong backbone networks.
> - For the computer vision task, a large dataset is needed to validate the proposed optimizer.
> - I'm not clear why this work cannot be applied to Riemannian networks, where most of the parameters lie in the manifold.
>
> Based on the above, I will keep the original score.

---

### Official Review · Reviewer_krcP · 2024-11-03

**Soundness:** 3
**Presentation:** 2
**Contribution:** 2
**Rating:** 5
**Confidence:** 4

**Summary:**

This paper introduces a Riemannian Jacobian Regularization technique to improve the generalization ability of deep models. The authors argue that by incorporating the Jacobian regularization on a Riemannian manifold, they can better capture the geometric structure of the parameter space, leading to improved performance in terms of generalization error. The theoretical analysis provides bounds on the generalization error, and empirical results demonstrate the effectiveness of the proposed method.

**Strengths:**

(1) This work tackles an important problem in deep learning, i.e., enhancing generalization, which is crucial for improving the models' robustness in real-world applications.

(2) The theoretical framework builds on well-established concepts in Riemannian geometry, providing a mathematically sound approach to regularization.

(3) The experimental results on different benchmarking datasets in supervised and self-supervised settings show the effectiveness of using the proposed RJR.

**Weaknesses:**

(1) Limited Theoretical Novelty: The theoretical contributions of this paper, e.g., Theorem 1, appear to rely on existing works (or experience) in the field of generalization theory, such as:
[a]. Hoffman, J., & Ma, Y. (2019). "Robust learning with Jacobian regularization." arXiv preprint arXiv:1907.05895.
[b]. Neyshabur, B., Tomioka, R., & Srebro, N. (2015). "Norm-based capacity control in neural networks." In Proceedings of the Conference on Learning Theory (COLT).
Hence, the lack of truly novel theoretical insights may weaken the originality of the paper. Addressing the limitations of current generalization bounds in the context of manifold-based regularization would be a meaningful addition.

(2) Methodological Contribution Needs Further Justification: Although the idea of Jacobian regularization on Riemannian manifolds is interesting, this paper lacks sufficient justification for why the proposed RJR would outperform or provide benefits compared to other regularization methods. Therefore, additional experiments comparing RJR to other regularization techniques (e.g., adversarial training, dropout) across a variety of tasks and architectures would help clarify its distinct advantages.

(3) Empirical Evaluation is limited: The experiments are conducted on a limited range of datasets and models. Expanding the empirical evaluation to include diverse datasets and model architectures would strengthen the paper. A key aspect is that this paper is about optimization on the Riemannian manifolds, but the authors do not applied the suggested RJR to Riemannian networks, such as SPDNet [c], SPDNetBN [d], GrNet [e], RResNet [f]. Additionally, it would be helpful to include ablation studies to show the sensitivity of the method to key hyperparameters, such as the choice of Riemannian manifolds, $\epsilon$, $\rho$.
[c] A riemannian network for spd matrix learning, AAAI, 2017.
[d] Riemannian batch normalization for SPD neural networks, NeurIPS, 2019
[e]  Building deep networks on grassmann manifolds, AAAI, 2018
[f] Riemannian residual neural networks, NeurIPS, 2023.

(4) Insufficient explanation: The role of the second term in $sigma_t$ (line 248) is unknown. In Section 5.2, the authors mentioned that enforcing orthogonality on the convolutional filters has some benefits, such as alleviating gradient vanishing. However, the basic reason is unknown, making the applicability of the proposed RJR unconvincing. Again, the authors are suggested to evaluate the effectiveness of RJR under end-to-end Riemannian networks when compared with RSGD and RSAM.

(5) Another limitation of this paper is that the English writing is obscure, especially in the proposed method section. In other word, it is  challenging for readers who are not already familiar with Riemannian geometry and Riemannian optimization.

(6) In line 356, $s$ means what?. In Fig. 2, why select $\lambda_5$. Figs. 2 and 3 are not vector graphics.

**Questions:**

Please refer to the weaknesses part for detailed information.

---

> ### Author Response · Authors · 2024-11-25
> **Response to Reviewer krcP (part 1)**
>
> We thank the reviewer for their constructive feedback. We would like to address the concerns as follows:
>
> + **Theoretical Contribution:** Firstly, we respectfully disagree with the comment of the reviewer that the main result relies on [1] and [2]. In particular, [1] focuses on improving model robustness with Jacobian regularization; while [2] shows the benefit of norm regularization to improving generalization, which is unrelated to the Jacobian. Our main theorem otherwise demonstrates the theoretical benefits of the *Jacobian* to *regularization*. Our analysis also builds upon and strengthens the generalization inequalities established in previous works on sharpness-aware minimization techniques like SAM [3] and FisherSAM [4]. To the best of our knowledge, this is the first generalization inequality for Riemannian manifolds. Moreover, a key novelty in this theoretical analysis is that the general loss is directly bounded by the empirical loss, instead of the worst-case empirical loss like prior results such as SAM [3] or FisherSAM [4]. Additionally, although Jacobian regularization techniques have been applied in the Euclidean setting in earlier works [1, 5], these methods lacked a theoretical foundation on the effect of Jacobian regularization on the generalization ability. Our work provides the first theoretical justification for this practice and extends it to the Riemannian context. Based on the reviewer's suggestion, we have incorporated this explanation in the theoretical development section of the rebuttal revision.
>
> + **Methodological Contribution**: The key feature of our algorithm that enhances its performance is the regularization of both the Riemannian gradient and the Jacobian. As demonstrated in our main theorem, this dual regularization effectively narrows the gap between the population loss $L_D$ and the empirical loss $L_S$, outperforming sharpness-aware methods that focus solely on implicitly regularizing the gradient norm. This is why we compare our method to Riemannian-SAM, which applies sharpness-aware minimization on manifolds, and SAM, a sharpness-aware technique in the Euclidean setting. Our results show that the sharpness in our models is significantly lower than in Riemannian-SAM, as confirmed in the ablation study, which in turn correlates with improved generalization performance.
>
> + **Regarding the empirical evaluations:** In response to the reviewer’s suggestion for more rigorous experiments on a broader range of datasets, we have conducted additional experiments on three more challenging datasets: Tiny-ImageNet, Caltech101, and CUB-200-2011. The results are presented below.
>
> |Methods |Tiny-Imagenet|Caltech101|CUB-200-2011|
> |-|-|-|-|
> |SGD|57.65|69.19|52.97|
> |SAM|60.27|70.38|54.72|
> |RSGD|59.34|69.02|53.31|
> |RSAM|61.55|71.46|56.22|
> |RJR|62.22|72.12|56.88|
>
> In Section 5, we have also conducted the ablation study to show the sensitivity to the key hyperparameters including $\epsilon$ and $\rho$. The results show that RJR is robust across reasonable ranges of these two hyperparameters.

---

> > ### Author Response · Authors · 2024-11-25
> > **Response to Reviewer krcP (part 2)**
> >
> > + **Regarding the other related explanations:**
> > - *The role of the second term in $\delta_t$:* As discussed in lines 234 - 237, the first term of $\delta_t$ implicitly regularizes the gradient norm. However, as mentioned earlier in lines 206 - 210, Lemma 1 and Theorem 1 suggest the need to also regularize $|grad_\theta(z\epsilon)|^2$ and $|\nabla_\theta(z\epsilon)|$ in addition to the gradient norm. Therefore, the second term of $\delta_t$ implicitly regularizes these two additional terms, which arise from the second-order term in Theorem 1.
> >
> > - **The Benefits of Orthogonal Convolutional Neural Networks (OCNNs):** OCNNs have been widely applied in a variety of domains, such as image generation and synthesis, where the orthogonal constraint encourages the network to explore a broader range of feature representations. In object detection, OCNNs help mitigate the risk of redundant feature learning. The primary motivation for enforcing orthogonality is that previous research has shown traditional convolutional networks often suffer from *representational collapse*, where filters in the deeper layers of the network learn similar or redundant features. By imposing orthogonality, these networks promote angular diversity among the filters, driving them to capture a more diverse set of features [6].
> >
> > + **Regarding other issues:** In line 356, $K_l$ refers to the matrix whose columns consist of the flattened $W_i$ matrices; this has been clarified in the revision. For the Hessian spectra appendix, we selected $\lambda_5$ because the ratio $\lambda_{\max}/\lambda_5$ is a widely used proxy for measuring sharpness and capturing the bulk of the spectrum, as discussed in prior literature [7]. This metric is commonly employed by SAM, which is why we also adopt it to ensure a fair comparison. We have revised the text to make these points clearer.
> >
> > Once again, we thank the reviewer for this constructive feedback.
> >
> > [1] Hoffman, J., & Ma, Y. (2019). "Robust learning with Jacobian regularization." arXiv preprint arXiv:1907.05895
> >
> > [2] Neyshabur, B., Tomioka, R., & Srebro, N. (2015). "Norm-based capacity control in neural networks." In Proceedings of the Conference on Learning Theory (COLT).
> >
> > [3] Foret, P., Malkin, N., & Cohn, T. (2021). Sharpness-aware minimization for efficiently improving generalization. Proceedings of the 38th International Conference on Machine Learning (ICML 2021), 26, 301-311.
> >
> > [4] Kim, M., Li, D., Hu, S. X., & Hospedales, T. (2022). Fisher SAM: Information geometry and sharpness aware minimization. Proceedings of the 39th International Conference on Machine Learning, 162, 11148–11161. PMLR. https://proceedings.mlr.press/v162/kim22f.html​
> >
> > [5] Sungyoon Lee, Jinseong Park, and Jaewook Lee. Implicit Jacobian regularization weighted with impurity of probability output. In Andreas Krause, Emma Brunskill, Kyunghyun Cho, Barbara Engelhardt, Sivan Sabato, and Jonathan Scarlett (eds.), Proceedings of the 40th International Conference on Machine Learning, volume 202 of Proceedings of Machine Learning Research, pp.19141–19184. PMLR, 23–29 Jul 2023
> >
> > [6] Jiayun Wang, Yubei Chen, Rudrasis Chakraborty, and Stella X. Yu. Orthogonal convolutional neural networks, 2020.
> >
> > [7] Stanislaw Jastrzebski, Maciej Szymczak, Stanislav Fort, Devansh Arpit, Jacek Tabor, Kyunghyun Cho, and Krzysztof Geras. The Break-Even Point on Optimization Trajectories of Deep Neural Networks. arXiv e-prints, art. arXiv:2002.09572, February 2020.

---

> > > ### Comment · Reviewer_krcP · 2024-12-01
> > >
> > > Thanks for the detailed responses provided by the authors, and some of my previous concerns have been addressed. However, the following issues are still unsolved:
> > >
> > > 1) The authors perform Riemannian optimization on the Stiefel manifold, but this needs to assume or restrict that the parameter matrix $U$ is semi-orthogonal. In this case, the authors should also make an ablation study to investigate the impact of matrix manifolds selection on the performance of your proposed method;
> > > 2) The authors just select some Euclidean backbones for comparison, limiting the scalability and adaptability of the proposed algorithm. Since the proposed RJR can be treated as an improved optimization algorithm on the Riemannian manifolds, the authors are suggested to evaluate your method on several pure end-to-end Riemannian backbones, as listed in my previous
> > > comments.
> > >
> > > Although RJR is an effective method as evidenced  in the experiments, some key points are neglected as mentioned above. This will limits the applicability of the proposed method. Therefore, I will keep my score unchanged. Hope the authors can consider these suggestions and make more comprehensive revisions.

---

### Official Review · Reviewer_JyZh · 2024-11-08

**Soundness:** 3
**Presentation:** 3
**Contribution:** 3
**Rating:** 6
**Confidence:** 4

**Summary:**

This paper introduces an optimization approach called Riemannian Jacobian Regularization for improving model generalization and robustness in constrained optimization problems. The authors first provide theoretical analysis showing how the population loss relates to empirical loss through Riemannian gradients and projected Hessians on manifolds. Based on this analysis, they develop the RJR algorithm which explicitly regularizes both the Riemannian gradient norm and the Jacobian while optimizing on manifolds. The authors demonstrate RJR's effectiveness across multiple tasks including supervised learning, labeled self-supervised learning, and unlabeled self-supervised learning, testing on various datasets and model architectures.

**Strengths:**

The paper considers both theoretical justifications as well as algorithms motivated from the theoretical development. A number of experiments have been provided to justify to proposed algorithm.

**Weaknesses:**

Major comment:



1. The theorem 2 of the paper of Foret et al 2021b seems to be a Euclidean version of the Theorem 1 of this paper. Instead of having norms of gradients and hessians of the loss function, they have max_(|eps| &lt; rho) L_S(theta+eps), i.e., the maximum empirical loss over a rho-neighborhood of the given parameter theta. One can imagine that if L_S is (locally) smooth with respect to theta then this term can be further bounded by norms of gradients and hessians using Taylor expansion arguments. On the other hand, L_S could have been non-smooth, e.g., it can be locally oscillating heavily but with small magnitudes, leading to Foret’s term still be bounded while the norms of gradients and hessians can explode. Do I miss something? If not, what are the benefits of working with a setting that needs a stronger assumption on local smoothness?
2. What do you mean by robustness? A priori one could think about robustness against adversarial pertubations in the input, distribution shift, randomness in the optimization dynamics, to name a few. Further, what set of experiments demonstrate the claim of “improving robustness”?

Minor comments:



1. The K-lipschitz assumption is in the restated version of Theorem 1 in the appendix but missing in the one in the main body.
2. Table 1 is somewhat hard to read. Perhaps keeping three significant figures would allow one to make the fonts larger - at any case, an accuracy 90.01 is not so different from 90.02.  Also, what are the numbers in parentheses in the last row? I presume it is the standard variance, but neither the caption nor the text in line 380 describes it.

**Questions:**

N/A

---

> ### Author Response · Authors · 2024-11-25
> **Response to Reviewer JyZh**
>
> Firstly, we would like to thank the reviewer for their valuable feedback. We address the concerns raised as follows:
>
> + **Regarding the main theorem:** As noted after the theorem statement, our result extends prior work on generalization, including that of Foret et al. (2021) [1]. There are two key distinctions in our approach. First, we generalize the analysis from the Euclidean setting to a Riemannian manifold. Second, as the reviewer highlighted, we directly relate the general loss $L_D$ to the empirical loss $L_S$, rather than using the worst-case empirical loss within a neighborhood (i.e., $\max_{|\theta' - \theta| < \epsilon} L_S(\theta')$). This formulation leads to a more straightforward optimization procedure. We also emphasize even though under the local smoothness assumption, the term $\max_{|\theta' - \theta| < \epsilon} L_S(\theta')$ can be related to $ L_S(\theta)$, our theorem does not rely on this assumption, since the derivation of our theorem is not a direct corollary of Foret et al. (2021).
>
> + **Regarding model robustness:** As shown in previous studies on sharpness-aware minimization, such as those by Foret et al. (2021) [1] and Kim et al. (2022) [2], a key benefit of sharpness-aware methods is their robustness to parameter perturbations. To support this claim for RJR, we conducted an additional ablation study, included in Appendix A.3 of the rebuttal revision, which assesses the robustness of RJR against adversarial parameter perturbations. The results show that RJR performs better than both Riemannian SAM and SGD in terms of model robustness.
>
> + **Regarding other minor issues:** The last row of the first table now correctly represents the 95% confidence interval for RJR. Additionally, we have addressed other minor issues raised by the reviewer in the revision.
>
> Once again, we would like to thank the reviewer for this valuable feedback.
>
> **References**:
>
> [1] Foret, P., Malkin, N., & Cohn, T. (2021). Sharpness-aware minimization for efficiently improving generalization. Proceedings of the 38th International Conference on Machine Learning (ICML 2021), 26, 301-311.
>
> [2] Kim, M., Li, D., Hu, S. X., & Hospedales, T. (2022). Fisher SAM: Information geometry and sharpness aware minimization. Proceedings of the 39th International Conference on Machine Learning, 162, 11148–11161. PMLR. https://proceedings.mlr.press/v162/kim22f.html​

---

### Author Response · Authors · 2024-11-25

We sincerely thank the reviewers for their valuable feedback. In response, we have prepared a revision with the following key updates:

- Based on the suggestions of the reviewers, we have included a more detailed explanation of the main theorem.

- In response to Reviewer JyZh, we have included another ablation study to assess the robustness of RJR to adversarial parameter perturbation.

- We have also addressed other minor issues suggested by all the reviewers.

We believe these revisions address the reviewers’ concerns and significantly improve the quality and clarity of the manuscript. Thank you for your thoughtful contributions.

---

### Meta-Review · Area_Chair_kHMb · 2024-12-19

**Metareview:**

The paper receives 4 informative reviews, 3 of which suggest rejecting the paper. In particular, Reviewer JyZh, Reviewer krcP and Reviewer ynnr point out that the theoretical novelty/contribution of the paper is not enough, Reviewer krcP, Reviewer Rjrt and Reviewer ynnr indicate that the empirical validations are unconvincing，and Reviewer ynnr considers the overall contributions as being quite limited. The author rebuttal fails to resolve the above main issues, thus I would recommend rejecting the paper.

**Additional Comments On Reviewer Discussion:**

As specified by Reviewer ynnr and Reviewer Rjrt, their main concerns are not addressed properly. For more details, please refer to the author-reviewer discussion form. In addition, I find the authors frequently evade the reviewers’ concerns. For example, the question of “What do you mean by robustness?” from Reviewer JyZh, who originally suggests weak acceptance but hasn’t response to the authors’ rebuttal, is not answered directly.

---

### Decision · Program_Chairs · 2025-01-22

Reject